# Who Leads and Who Follows in Strategic Classification?

**Tijana Zrnic**\*
University of California, Berkeley
tijana.zrnic@berkeley.edu

**Eric Mazumdar**\*
California Institute of Technology
mazumdar@caltech.edu

**S. Shankar Sastry**
University of California, Berkeley
sastry@eecs.berkeley.edu

**Michael I. Jordan**
University of California, Berkeley
jordan@cs.berkeley.edu

## Abstract

As predictive models are deployed into the real world, they must increasingly contend with strategic behavior. A growing body of work on strategic classification treats this problem as a *Stackelberg game*: the decision-maker "leads" in the game by deploying a model, and the strategic agents "follow" by playing their best response to the deployed model. Importantly, in this framing, the burden of learning is placed solely on the decision-maker, while the agents' best responses are implicitly treated as instantaneous. In this work, we argue that the order of play in strategic classification is fundamentally determined by the relative frequencies at which the decision-maker and the agents adapt to each other's actions. In particular, by generalizing the standard model to allow *both* players to learn over time, we show that a decision-maker that makes updates faster than the agents can reverse the order of play, meaning that the agents lead and the decision-maker follows. We observe in standard learning settings that such a role reversal can be desirable for both the decision-maker and the strategic agents. Finally, we show that a decision-maker with the freedom to choose their update frequency can induce learning dynamics that converge to Stackelberg equilibria with either order of play.

## 1 Introduction

Individuals interacting with a decision-making algorithm often adapt strategically to the decision rule in order to achieve a desirable outcome. While such strategic adaptation might increase the individuals' utility, it also breaks the statistical patterns that justify the decision rule's deployment. This widespread phenomenon, often known as Goodhart's law, can be summarized as: "When a measure becomes a target, it ceases to be a good measure" [49].

A growing body of work known as *strategic classification* [14, 19, 28] models this phenomenon as a two-player game in which a decision-maker "leads" and strategic agents subsequently "follow." Specifically, the decision-maker first deploys a decision rule, and the agents then take a strategic action so as to optimize their outcome according to the deployed rule, subject to natural manipulation costs. For example, a bank might make lending decisions using applicants' credit scores. Knowing this mechanism, loan applicants might sign up for a large number of credit cards in an effort to strategically increase their credit score at little effort.

One of the main goals in the literature is to develop strategy-robust decision rules; that is, rules that remain meaningful even after the agents have adapted to them. Recent work has studied strategies for

---

\*Equal contribution.

35th Conference on Neural Information Processing Systems (NeurIPS 2021).

finding such rules through *repeated interactions* between the decision-maker and the agents [6, 20, 46]. In particular, the decision-maker sequentially deploys different rules, and for each they observe the population's response. Under certain regularity conditions, over time the decision-maker can find the optimal solution, defined as the rule that minimizes the decision-maker's loss *after* the agents have responded to the rule.

With the emergence of online platforms such as social media and e-commerce sites, repeated interactions between decision-makers and the population have become ever more prevalent. Online platforms continuously monitor user behavior and update pricing algorithms, recommendation systems, and popularity rankings accordingly. Users, on the other hand, take actions to ensure favorable outcomes in the face of these updates.

A distinctive feature of online platforms is the decision-maker's dominant computational power and abundant data resources, allowing the platform to react to any change in the agents' behavior virtually instantaneously. For example, if fake news content changes over time, automated algorithms can quickly detect this and retrain the classifier to incorporate the shift. It has been observed [see, e.g., 16, 18, 45] that, when faced with such "reactive" algorithms, strategic agents tend to take actions that *anticipate* the algorithm's response. That is, through repeated interactions, agents aim to find actions that maximize the agents' utility *after* the decision-maker has responded to these actions. This suggests that the order of play in strategic interactions can in fact be *reversed*, such that the agents "lead" while the decision-maker "follows."

To give an example of such a reversed strategic interaction, consider ride-sharing platforms that deploy algorithms for determining travel fare as a function of trip length and relevant traffic conditions. These pricing mechanisms are frequently updated based on the current supply and demand, and in particular a dip in the supply of drivers triggers a surge pricing algorithm. Möhlmann and Zalmanson [45] observed that drivers occasionally coordinate a massive deactivation of drivers from the system, artificially lowering driver supply, only to get back on the platform after some time has passed and the prices have surged. In this interaction, *the drivers essentially make the first move*, while the platform's pricing algorithm reacts to their action. Other examples of users aiming to exert control over algorithms can be found in the context of social network analyses [16, 18].

In this work, we argue that the order of play in strategic classification is fundamentally tied to the relative *update frequencies* at which the decision-maker and the strategic agents adapt to each other's actions. In particular, we show that, by tuning their update frequency appropriately, the decision-maker can select the order of play in the underlying game. Furthermore, in natural settings we show that allowing the strategic agents to play first in the game can actually be preferable for *both* the decision-maker and the agents. This is contrary to the order of play previously studied in the literature, whereby the decision-maker is always assumed to make the first move.

## 1.1   Our contribution

To give an overview of our results, we recall some relevant game-theoretic concepts. In the existing literature strategic classification is modeled as a *Stackelberg game*. A Stackelberg game is a two-person game where one player, called the *leader*, moves first, and the other player, called the *follower*, moves second, with the possibility of adapting to the move of the leader. Previous work assumes that the decision-maker acts as the leader and the agents act as the follower. This means that the decision-maker first deploys a model, and the agents then modify their features at some cost in order to obtain a favorable outcome according to the model. The decision-maker's goal is to find the *Stackelberg equilibrium*—the model that minimizes the decision-maker's loss after the agents have optimally adapted to it. This optimal reaction by the agents is called the *best response* to the model.

An important parameter that has been largely overlooked in existing work is the rate at which the agents re-evaluate and potentially modify their features. Most works studying the interaction between a decision-maker and strategic agents implicitly assume that, as soon as the model is updated, the data collected from strategic agents follows the best response to the currently deployed model. In the current work we do *not* assume that the agents react instantaneously to model updates. Instead, we assume that there is a natural timescale according to which the agents adapt their features to models.

Allowing agents to adapt gradually to deployed models gives the decision-maker a new dimension upon which to act strategically. Faced with agents that adapt gradually, the decision-maker can *choose* the timescale at which they update the deployed model. In particular, they can choose a rate

of updates that is *faster* than the agents' rate, or they can choose a rate that is *slower* than the agents' rate. We call decision-makers that follow a faster clock than the agents *reactive*, and if they follow a slower clock we call them *proactive*. Given that existing work on strategic classification relies on instantaneous agent responses, the previously studied decision-makers are all implicitly proactive.

Our first main result states that the decision-maker's choice of whether to be proactive or reactive fundamentally determines the order of play in strategic classification. Perhaps counterintuitively, by choosing to be *reactive* it is possible for the decision-maker to let the *agents* become the leader in the underlying Stackelberg game. Since changing the order of play changes the game's natural equilibrium concept, this choice can have a potentially important impact on the solution that the decision-maker and agents find. Throughout, we refer to the Stackelberg equilibrium when the decision-maker leads as the *decision-maker's equilibrium* and the Stackelberg equilibrium when the agents lead as the *strategic agents' equilibrium*.

**Theorem 1.1** (Informal). *If the decision-maker is proactive, the natural dynamics of strategic classification converge to the decision-maker's equilibrium. If the decision-maker is reactive, the natural dynamics of strategic classification converge to the strategic agents' equilibrium.*

To provide some intuition for Theorem 1.1, imagine that one player makes updates with far greater frequency than the other player. This allows the faster player to essentially converge to their best response between any two updates of the slower player. The slower player is then faced with a Stackelberg problem: they have to choose an action, expecting that the faster player will react optimally after their update. As a result, the optimal choice for the slower player is to drive the dynamics toward the Stackelberg equilibrium where they act as the leader.

It is well known (see, e.g., Section 4.5 in [5]) that under general losses, either player can prefer to lead or follow in a Stackelberg game, meaning that a player achieves lower loss at the corresponding equilibrium. Our second main takeaway is that in classic learning problems it can be preferable for *both* the decision-maker and the agents if the agents lead in the game and the decision-maker follows. One setting where this phenomenon arises is logistic regression with static labels and manipulable features.

**Theorem 1.2** (Informal). *Suppose that the decision-maker implements a logistic regression model and the strategic agents aim to maximize their predicted outcome. Then, both the decision-maker and the strategic agents prefer the strategic agents' equilibrium to the decision-maker's equilibrium.*

Theorem 1.2 suggests that there are other meaningful equilibria than those previously studied in the literature. Moreover, Theorem 1.1 proves that such equilibria can naturally be achieved if the decision-maker is *reactive* and agents are no-regret. Seeing that the decision-maker's equilibrium has also been shown to imply a cost to social welfare [30, 44], our results pave the way for studying new, potentially more desirable solutions in strategic settings.

## 1.2 Related work

Our work builds on the growing literature on strategic classification [see, e.g., 2, 11, 14, 19, 20, 25, 26, 28, 30, 35, 38, 42, 44, 51, and the references therein]. In these works, a decision-maker seeks to deploy a predictive model in an environment where strategic agents attempt to respond in a *post hoc* manner to maximize their utility given the model. Given this framework, a number of recent works have studied natural learning dynamics for learning models that are robust to strategic manipulation of the data [6, 17, 20, 30, 38, 48]. Such problems have also been studied in the more general setting of performative prediction [13, 31, 41, 43, 46]. Notably, all of these works model the interaction between the decision-maker and the agents as a repeated Stackelberg game [50] in which the decision-maker leads and the agents follow, and these roles are immutable. This follows in a long and continuing line of work in game theory on playing in games with hierarchies of play [3–5].

Learning in Stackelberg games is itself a growing area in game-theoretic machine learning. Recent work has analyzed the asymptotic convergence of gradient-based learning algorithms to local notions of Stackelberg equilibria [20–22] assuming a fixed order of play. The emphasis has largely been on zero-sum Stackelberg games, due to their structure and relevance for min-max optimization and adversarial learning [21, 33]. Such results often build upon work on two-timescale stochastic approximations [10, 36] in which tools from dynamical systems theory are used to analyze the limiting behavior of coupled stochastically perturbed dynamical systems evolving on different timescales.

In this paper we depart from this prior work in both our problem formulation and our analysis of learning algorithms. To begin, the problem we consider is asymmetric: one player, namely the strategic agents, makes updates at a fixed frequency, while the opposing player, the decision-maker, can strategically choose their update frequency as a function of the equilibrium to which they wish to converge. Thus, unlike prior work, the choice of timescale becomes a strategic choice on the part of the decision-maker and consequently the order of play in the Stackelberg game is not predetermined.

Our analysis of learning algorithms is also more involved than previous works since *both* the leader and follower make use of learning algorithms. Indeed, throughout our paper we assume that the population of agents is using *no-regret learning* algorithms, common in economics and decision theory [8, 9, 24, 27, 29, 39, 47]. This captures the reality that agents *gradually* adapt to the decision-maker's actions on some natural timescale. In contrast, existing literature on strategic classification and learning in Stackelberg games assume that the strategic agents or followers are always playing their best response to the leader's action. One exception is recent work [32] that replaces agents' best responses with *noisy* responses, which are best responses to a perturbed version of the deployed model. While this model does capture imperfect agent behavior, it does not address the effect of learning dynamics and relative update frequencies on the agents' actions.

Given this assumption on the agents' learning rules, we then show how decision-makers who reason strategically about their relative update frequency can use simple learning algorithms and still be guaranteed to converge to *game-theoretically meaningful* equilibria. Our analysis complements a line of recent work on understanding gradient-based learning in continuous games, but crucially does not assume that the two players play simultaneously [see, e.g., 12, 40]. Instead, we analyze cases where *both* the agents and decision-maker learn over time, and play asynchronously.

This represents a departure from work in economics on understanding equilibrium strategies in games and the benefits of different orders of play [see, e.g., 3, 4] in Stackelberg games. In our problem formulation, players do not have fixed strategies but must learn a strategy by learning about both their loss and their opponent's through repeated interaction.

Some of our analyses touch on ideas from online convex optimization [52], specifically derivative-free optimization [1, 15, 23, 37]. Several works [20, 43] within strategic classification and performative prediction apply similar zeroth-order tools to find the decision-maker's equilibrium, but once again assuming immediate best responses to deployed models. We show that such algorithms are versatile enough to be used without such strong assumptions while still having strong convergence guarantees.

## 1.3  Organization

This paper is organized as follows. In Section 2 we introduce our model for studying the interaction between a decision-maker and strategic agents that adapt gradually to deployed decision rules, and formalize the concept of *reactive* and *proactive* decision-makers. In Section 3 we show that under natural assumptions, a proactive or reactive decision-maker can drive the game towards the decision-maker's or agents' equilibrium, respectively, by using simple learning rules. We follow this in Section 4 by showing how in simple learning problems inducing a certain order of play can benefit *both* the decision-maker and the agents. We conclude in Section 5 with a brief discussion of the questions our proposed model raises and some directions for future work.

## 2  Model

We start with an overview of the basic concepts and notation, and then discuss the main conceptual novelty of our work—implications of the decision-maker's and strategic agents' update frequencies.

## 2.1  Basic concepts and notation

Throughout we denote by $z = (x, y)$ the (feature, label) pairs corresponding to the strategic agents' data. We assume that the decision-maker chooses a model parameterized by $\theta \in \Theta \subseteq \mathbb{R}^d$, where $\Theta$ is convex and closed, and that their loss is measured via a convex loss function $\ell(z; \theta)$. The strategic agents measure loss via a function $r(z; \theta)$ and, collectively, they form a distribution in the family $\{\mathcal{P}(\mu) : \mu \in \mathcal{M} \subseteq \mathbb{R}^m\}$, where $\mathcal{M}$ is convex and closed. Here, $\mu$ denotes the aggregate summary of

all agents' actions. The data observed by the decision-maker is $\mathcal{P}(\mu)$ and as such varies depending on the agents' aggregate action $\mu$.

We denote $L(\mu, \theta) = \mathbb{E}_{z \sim \mathcal{P}(\mu)} \ell(z; \theta)$, and $R(\mu, \theta) = \mathbb{E}_{z \sim \mathcal{P}(\mu)} r(z; \theta)$. With this, the agents' best response is given by $\mu_{\mathrm{BR}}(\theta) = \arg\min_\mu R(\mu, \theta)$ and the decision-maker's best response is given by $\theta_{\mathrm{BR}}(\mu) = \arg\min_\theta L(\mu, \theta)$. We assume that the best responses for both players are always unique.

If the decision-maker acts as the leader in the game, their incurred *Stackelberg risk* is equal to $\mathrm{SR}_L(\theta) = L(\mu_{\mathrm{BR}}(\theta), \theta)$. Similarly, we let $\mathrm{SR}_R(\mu) = R(\mu, \theta_{\mathrm{BR}}(\mu))$ denote the Stackelberg risk of the agents when they lead in the game. We let $\theta_{\mathrm{SE}}$ and $\mu_{\mathrm{SE}}$ denote the decision-maker's and strategic agents' equilibrium, respectively: $\theta_{\mathrm{SE}} = \arg\min_\theta \mathrm{SR}_L(\theta)$ and $\mu_{\mathrm{SE}} = \arg\min_\mu \mathrm{SR}_R(\mu)$. We assume that each equilibrium is unique. Note that the two players cannot compute their respective equilibrium "offline", as we do not assume they have access to the other player's loss function.

As discussed earlier, we assume that there is an underlying timescale according to which the agents re-evaluate their features. Specifically, after each time interval of fixed length, the agents observe the currently deployed model, as well as their loss according to that model, and possibly modify their features accordingly. The decision-maker, aware of the agents' timescale, can choose to be *proactive*, meaning they choose an update frequency slower than that of the agents, or *reactive*, meaning they choose a higher update frequency. This power asymmetry that allows the decision-maker to choose a timescale is characteristic of online platforms with abundant resources. Implicit in our setup in an assumption that the decision-maker can evaluate the agents' update frequency and adapt to it. Big IT companies generally employ monitoring systems that detect distribution shift; at a high level, the rate of distribution shift can be thought of as the rate of agents' adaptation.

We use the term *epoch* to refer to a period between two updates of the *slower* player (which player is the slower one is up to the decision-maker). In particular, the $t$-th epoch starts with a single update of the slower player, followed by $\tau \in \mathbb{N}$ updates of the faster player. The rate $\tau$ is fixed.

We use $\theta_t$ and $\mu_t$ to denote the iterate of the decision-maker and the strategic agents, respectively, *at the end* of epoch $t$. Furthermore, for the faster player, we use double-indexing to denote the within-epoch iterates. For example, if the decision-maker is the faster player, we use $\{\theta_{t,j}\}_{j=1}^\tau$ to denote their iterates within epoch $t$. Note that $\theta_{t,\tau} \equiv \theta_t$. We also let $\bar{\theta}_t = \frac{1}{\tau} \sum_{j=1}^\tau \theta_{t,j}$. We adopt similar notation when the agents have a higher update frequency.

## 2.2 Rational agents in the face of varying update frequencies

Adopting the distinction between reactive and proactive decision-makers, it is crucial to re-evaluate what it means for the strategic agents to behave rationally. We argue that rational behavior must depend on the relative update frequencies of the decision-maker and the agents.

As a running toy example, consider a decision-maker building a model with the goal of distinguishing between spam and legitimate emails. The population of strategic agents aims to craft emails that bypass the decision-maker's spam filter. Here, $\mu$ could determine the number of words in an email, types of words used, etc. The loss $R(\mu, \theta)$ could be some decreasing function of the number of daily clicks on email content, given spam filter $\theta$ and emails crafted according to $\mu$. In the following discussion assume that the timescales of the decision-maker and the agents have a significant separation: the decision-maker is either "significantly faster" or "significantly slower." As we will make more formal later on, our results will generally assume a sufficiently large separation between the timescales. In the following paragraphs we informally describe rational agent behavior in the context of update frequencies.

**Proactive decision-maker.** First, assume that the decision-maker is proactive, and suppose they deploy model $\theta$. By definition, this model remains in place for a relatively long time, as observed by the agents. Then, by choosing features $\mu$, the agents experience loss $R(\mu, \theta)$ during that period, and as a result the most rational decision is to choose features $\mu_{\mathrm{BR}}(\theta)$. In the running example, if $\theta$ is a spam filter that is in place for many months, it is rational for spammers to craft emails that are most likely to bypass filter $\theta$. This is just the usual best response—as we alluded to earlier, when the decision-maker is proactive, our setup is similar to that of previous work.

**Reactive decision-maker.** Now assume that the decision-maker is reactive, and suppose the agents observe $\theta$ as the current model. Then, by setting $\mu$, the agents do *not* experience loss $R(\mu, \theta)$. Rather,

their loss is $R(\mu, \theta_R(\mu))$, where $\theta_R(\mu)$ denotes the decision-maker's *reaction* to the agents' choice $\mu$. In the spam example, suppose that the decision-maker can aggregate and process data quickly, and retrains the spam filter every couple of hours. Moreover, suppose that the spammers adapt their emails only once per week. Then, the agents' loss after choosing $\mu$ (evaluated weekly) is determined by the number of clicks allowed by the updated filter $\theta_R(\mu)$, *not* the old filter $\theta$. Therefore, if the agents could predict $\theta_R(\mu)$, the agents' optimal decision would be to choose $\arg\min_\mu R(\mu, \theta_R(\mu))$. In other words, rather than choose the best response to $\theta$, rational agents interacting with a reactive decision-maker would choose $\mu$ so that it triggers the *best possible reaction* from $\theta$.

We formalize this intuitive behavior by assuming that the agents are *no-regret* learners [47]. This essentially means that their average regret vanishes as the number of actions grows. More formally, we assume the following behavior depending on the relative update frequencies:

- If the decision-maker is proactive, then for any $\theta_t$, the agents' strategy ensures:

$$\frac{1}{\tau} \sum_{j=1}^\tau \mathbb{E} R(\mu_{t,j}, \theta_t) - \min_\mu R(\mu, \theta_t) \to 0 \text{ as } \tau \to \infty. \tag{A1}$$

- If the decision-maker is reactive, then for any response function $\theta_R(\mu)$, the agents' strategy ensures:

$$\frac{1}{T} \sum_{t=1}^T \mathbb{E} R(\mu_t, \theta_R(\mu_t)) - \min_\mu R(\mu, \theta_R(\mu)) \to 0 \text{ as } T \to \infty, \tag{A2}$$

whenever such a strategy exists. If the agents' loss is convex, the first condition can be satisfied by simple gradient descent. In fact, gradient descent would typically imply an even stronger guarantee, namely the convergence of the iterates, $\mu_{t,\tau} \to \mu_{BR}(\theta_t)$. The second condition can be satisfied by various bandit strategies if $R(\mu, \theta_R(\mu))$ is Lipschitz and $\mathcal{M}$ is bounded (and we will impose these conditions explicitly in the following section). That said, it seems hardly suitable to assume that the agents run a well-specified optimization procedure. For this reason, we will for the most part avoid making explicit algorithmic assumptions on the agents' strategy and our main takeaways will only rely on rational agent behavior in the limit, as in equations (A1) and (A2).

## 3 Learning dynamics

In this section, we study the limiting behavior of the interaction between the decision-maker and the strategic agents. We show that, by running classical optimization algorithms, the decision-maker can drive the interaction to a Stackelberg equilibrium with either player acting as the leader.

### 3.1 Convergence to decision-maker's equilibrium

In general, we do not expect the decision-maker to be able to compute derivatives of the function $SR_L$. For this reason, to achieve convergence to the decision-maker's equilibrium, we consider running a derivative-free method. One such solution is the "gradient descent without a gradient" algorithm of Flaxman et al. [23]. Past work [20, 43] also considers this algorithm with the goal of optimizing $SR_L$, but it assumes instantaneous agent responses. In other words, it assumes query access to $SR_L$ directly, while we consider perturbations due to imperfect agent responses. It is worth noting that, under further assumptions, one could apply more efficient two-stage approaches [31, 43] that approximate the gradients of $SR_L$ by first estimating the best-response map $\mu_{BR}$.

Specifically, we let the decision-maker run the following update:

$$\phi_{t+1} = \Pi_\Theta(\phi_t - \eta_t \frac{d}{\delta} L(\bar{\mu}_t, \phi_t + \delta u_t) u_t), \text{ where } u_t \sim \text{Unif}\left(\mathcal{S}^{d-1}\right). \tag{1}$$

Here, $\Pi_\Theta$ denotes the Euclidean projection, $\text{Unif}\left(\mathcal{S}^{d-1}\right)$ denotes the uniform distribution on the unit sphere in $\mathbb{R}^d$, $\eta_t$ is a non-increasing step size sequence, and $\delta > 0$ is a fixed hyperparameter. The deployed model in the $t$-th epoch is set as $\theta_t = \phi_t + \delta u_t$.[2]

---

[2]Technically, this assumes that we can deploy a model in a $\delta$-ball around $\Theta$. Another solution would be to use a projection onto a small contraction of $\Theta$ in equation (1). This is a minor technical hurdle common in the literature. The rate in Theorem 3.1 is unaffected by the choice of solution to this technical point.

We provide convergence guarantees assuming that the decision-maker's Stackelberg risk $\mathrm{SR}_L$ is convex. While this condition doesn't follow from convexity of the loss $\ell(z;\theta)$ alone, previous work has established conditions for convexity of this objective for different learning problems and agent utilities [20, 43]. For example, in the linear and logistic regression examples discussed in the following section, the decision-maker's Stackelberg risk will be convex.

**Theorem 3.1.** *Denote by $D_\Theta$ the diameter of $\Theta$, and suppose that $|L(\mu, \theta)| \leq B$ for all $\mu, \theta$. Furthermore, suppose that $\mathrm{SR}_L$ is convex and $\beta$-Lipschitz and $L(\mu, \theta)$ is $\beta_\mu$-Lipschitz in the first entry for all $\theta$. Then, if the decision-maker runs update* (1) *with $\eta_t = \eta_0 d^{-\frac{1}{2}} t^{-\frac{3}{4}}$ and $\delta = \delta_0 d^{\frac{1}{2}} T^{-1/4}$, it holds that*

$$\sum_{t=1}^{T}(\mathbb{E}[\mathrm{SR}_L(\theta_t)] - \mathrm{SR}_L(\theta_{\mathrm{SE}})) \leq \left(\frac{D_\Theta^2}{2\eta_0} + \frac{2B^2}{\delta_0^2}\right)\sqrt{d}T^{3/4} + \beta_\mu D_\Theta \sum_{t=1}^{T}\mathbb{E}\|\bar{\mu}_t - \mu_{\mathrm{BR}}(\theta_t)\|_2.$$

*Moreover, assuming that the agents are rational* (A1) *and $\mathcal{M}$ is compact, we have*

$$\lim_{\tau \to \infty}\sum_{t=1}^{T}(\mathbb{E}[\mathrm{SR}_L(\theta_t)] - \mathrm{SR}_L(\theta_{\mathrm{SE}})) \leq \left(\frac{D_\Theta^2}{2\eta_0} + \frac{2B^2}{\delta_0^2}\right)\sqrt{d}T^{3/4}. \tag{2}$$

**Remark 3.2.** *For Theorem 3.1, we assume that the agents are rational in a relatively weak sense, by assuming no-regret behavior. Often, however, we expect the agents' strategy to achieve* iterate convergence, *and not just vanishing regret. More precisely, it makes sense to expect $\mu_{t,\tau} \to \mu_{\mathrm{BR}}(\theta_t)$ as $\tau \to \infty$. For example, this guarantee is achieved by gradient descent in a variety of settings. In that case, the decision-maker can simply use the* last *iterate instead of the average one:*

$$\phi_{t+1} = \Pi_\Theta(\phi_t - \eta_t \frac{d}{\delta} L(\mu_t, \phi_t + \delta u_t)u_t), \text{ where } u_t \sim \mathrm{Unif}\left(\mathcal{S}^{d-1}\right). \tag{3}$$

*Similarly, $\mathbb{E}\|\bar{\mu}_t - \mu_{\mathrm{BR}}(\theta_t)\|_2$ would be replaced by $\mathbb{E}\|\mu_t - \mu_{\mathrm{BR}}(\theta_t)\|_2$ in the bound of Theorem 3.1.*

**Remark 3.3.** *The rate $O(\sqrt{d}T^{3/4})$ is characteristic of the Flaxman et al. [23] zeroth-order algorithm. Subsequent work on bandit convex optimization has improved upon this rate in terms of $T$ at the cost of worse dependence on $d$ [1, 15, 37]. In this work we opt to analyze the Flaxman et al. method given its simplicity and the fact that its rate has not been uniformly improved upon. That said, we do not anticipate any difficulties in proving a result analogous to Theorem 3.1 in the context of a different bandit convex optimization algorithm. Specifically, we only require that the error $\|\bar{\mu}_t - \mu_{\mathrm{BR}}(\theta_t)\|_2$ propagates "smoothly" to the decision-maker's regret.*

In some cases, the additional regret due to imperfect agent responses does not alter the asymptotic rate at which the decision-maker accumulates regret even if the epoch length $\tau$ is constant and does not grow with $T$. To illustrate this point, we consider strategic agents that follow the gradient-descent direction on a possibly nonconvex objective with enough curvature. More precisely, we assume that for all $\theta$, $R(\mu, \theta)$ satisfies the Polyak-Łojasiewicz (PL) condition:

$$\gamma(R(\mu, \theta) - \min_{\mu \in \mathcal{M}} R(\mu, \theta)) \leq \frac{1}{2}\|\nabla_\mu R(\mu, \theta)\|_2^2,$$

for some parameter $\gamma > 0$. Suppose that the agents' update is computed as:

$$\mu_{t,j+1} = \mu_{t,j} - \eta_\mu \nabla_\mu R(\mu_{t,j}, \theta_t), \tag{4}$$

where $\eta_\mu > 0$ is a constant step size and $\mu_{t,0} = \mu_{t-1,\tau}$. In this case, gradient descent achieves last-iterate convergence and hence we assume that the decision-maker uses the update in equation (3).

**Theorem 3.4.** *Assume the conditions of Theorem 3.1. In addition, suppose that $R(\mu, \theta)$ is $\beta_\mu^R$-smooth in $\mu$ for all $\theta$ and satisfies the PL condition with parameter $\gamma$, and $\mu_{\mathrm{BR}}(\theta)$ is $\beta_{\mathrm{BR}}$-Lipschitz in $\theta$. Assume that the strategic agents run update* (4) *with $\eta_\mu < \frac{1}{\beta_\mu^R}$. Further, suppose the epoch length is chosen so that $\tau > \log(\beta_\mu^R/\gamma)/\log(1/(1 - \gamma\eta_\mu))$. Then, for some constant $\alpha(\tau) \in (0, 1)$, we have*

$$\sum_{t=1}^{T}\mathbb{E}\|\mu_t - \mu_{\mathrm{BR}}(\theta_t)\|_2 \leq \frac{\|\mu_0 - \mu_{\mathrm{BR}}(\theta_0)\|_2 + \frac{4\beta_{\mathrm{BR}}B\eta_0}{\delta_0}\sqrt{T}}{1 - \alpha(\tau)}.$$

Therefore, the decision-maker's regret is $O(\sqrt{d}T^{3/4})$ even with a constant epoch length. This result crucially depends on the fact that the optimization problems that the agents solve in neighboring epochs are coupled through $\mu_{t,0} = \mu_{t-1,\tau}$. If $\mu_{t,0}$ were reinitialized arbitrarily in each epoch, the extra regret would be linear in $T$ given constant epoch length.

By using standard tools from the stochastic approximation literature [10], in the Appendix we additionally provide convergence to local optima for the update (3) when $\mathrm{SR}_L$ is possibly nonconvex.

## 3.2 Convergence to strategic agents' equilibrium

Now we analyze the case when the decision-maker is reactive. Given a large enough gap in update frequencies—that is, a large enough epoch length $\tau$—the decision-maker can converge to their best response to the current iterate $\mu_t$ between any two actions of the agents. The most natural choice for achieving this is to run standard gradient descent, $\theta_{t,k+1} = \theta_{t,k} - \eta_k \nabla_\theta L(\mu_t, \theta_{t,k})$. In what follows we provide asymptotic guarantees assuming that the decision-maker runs any algorithm that achieves iterate convergence. This condition can be satisfied by gradient descent in a variety of settings. Formally, we assume that for any fixed $\mu_t$, the decision-maker's strategy ensures

$$\|\theta_{t,\tau} - \theta_{\mathrm{BR}}(\mu_t)\|_2 \to_p 0, \tag{5}$$

as $\tau \to \infty$. Here, $\to_p$ denotes convergence in probability.

We first observe that, in the limit as $\tau$ grows, the agents' accumulated risk is equal to their accumulated *Stackelberg risk* at all the actions played so far. This simply follows by continuity.

**Lemma 3.5.** *Suppose that the decision-maker achieves iterate convergence* (5) *and $R$ is continuous in the second argument. Then, for all $T \in \mathbb{N}$, $\lim_{\tau \to \infty} \sum_{t=1}^T \mathbb{E}R(\mu_t, \theta_t) = \sum_{t=1}^T \mathbb{E}\mathrm{SR}_R(\mu_t)$.*

In other words, in every epoch the agents essentially play a Stackelberg game in which they lead and the decision-maker follows. This holds regardless of whether the agents behave rationally. If they do behave rationally (condition (A2)), we show that both the agents' and the decision-maker's average regret with respect to $(\mu_{\mathrm{SE}}, \theta_{\mathrm{BR}}(\mu_{\mathrm{SE}}))$ vanishes if the agents' updates are continuous. To formalize this, suppose that for all $t \in \mathbb{N}$, the agents set $\mu_{t+1} = D_{t+1}(\mu_1, \theta_1, \ldots, \mu_t, \theta_t, \xi_{t+1})$, where $D_{t+1}$ is some fixed map and $\xi_{t+1}$ is a random variable independent of $\{(\mu_i, \theta_i)\}_{i \le t}$. We include $\xi_{t+1}$ as an input to allow randomized strategies. Then, we will say that the agents' updates are *continuous* if $D_{t+1}$ is continuous in the first $2t$ coordinates for all $t \in \mathbb{N}$.

**Theorem 3.6.** *Suppose that the agents' updates are continuous and rational* (A2)*, and that $\mathcal{M}$ is compact. Further, suppose that the decision-maker achieves iterate convergence* (5) *and $\mathrm{SR}_R$ and $\mathrm{SR}_L$ are Lipschitz. Then, it holds that*

$$\lim_{T \to \infty} \lim_{\tau \to \infty} \frac{1}{T} \sum_{t=1}^T \mathbb{E}\mathrm{SR}_R(\mu_t) - \mathrm{SR}_R(\mu_{\mathrm{SE}}) = 0,$$

$$\lim_{T \to \infty} \lim_{\tau \to \infty} \frac{1}{T} \sum_{t=1}^T \mathbb{E}L(\mu_t, \theta_t) - L(\mu_{\mathrm{SE}}, \theta_{\mathrm{BR}}(\mu_{\mathrm{SE}})) = 0.$$

# 4 Preferred order of play

While we have shown that the decision-maker can tune their update frequency to achieve either order of play in the Stackelberg game, it remains to understand which order of play is preferable for the decision-maker and the strategic agents. In the following examples, we illustrate that in classic learning settings both players can prefer the order when the *agents lead*. This suggests that the natural and overall more desirable order of play is sometimes reversed compared to the order usually studied.

At first, it might seem counterintuitive that the decision-maker could prefer to follow. To get some intuition for why following might be preferred to leading, recall that in zero-sum games *following is never worse*. In particular, suppose $R(\mu, \theta) = -L(\mu, \theta)$. Then, the basic min-max inequality says

$$L(\mu_{\mathrm{SE}}, \theta_{\mathrm{BR}}(\mu_{\mathrm{SE}})) = \max_\mu \min_\theta L(\mu, \theta) \le \min_\theta \max_\mu L(\mu, \theta) = L(\mu_{\mathrm{BR}}(\theta_{\mathrm{SE}}), \theta_{\mathrm{SE}}),$$

with equality if and only if a *Nash* equilibrium exists. Therefore, if a Nash equilibrium does not exist, following is strictly preferred.

Since strategic classification is typically not a zero-sum game, we look at two common learning problems and analyze the preferred order of play.

## 4.1 Linear regression

Suppose that the agents' non-strategic data, $(x_0, y)$, where $x_0$ is a feature vector and $y$ the outcome, is generated according to

$$x_0 \sim \mathcal{P}_0, \ y = x_0^\top \beta + \xi,$$

where $\mathcal{P}_0$ is a zero-mean distribution such that $\mathbb{E}_{x_0 \sim \mathcal{P}_0} x_0 x_0^\top = I$, $\beta \in \mathbb{R}^d$ is an arbitrary fixed vector, and $\xi$ has mean zero and finite variance $\sigma^2$. We denote the joint distribution of $(x_0, y)$ by $\mathcal{P}(0)$.

Recall that we use $z$ to denote the pair $(x, y)$. Suppose that the decision-maker runs standard linear regression with the squared loss:

$$\ell(z; \theta) = \frac{1}{2}(y - x^\top \theta)^2.$$

The agents aim to maximize their predicted outcome, $r(z; \theta) = -\theta^\top x$, subject to a fixed budget on feature manipulation—they can move to any $x$ at distance at most $B$ from their original features $x_0$: $\|x - x_0\|_2 \le B$. A similar model is considered by Kleinberg and Raghavan [35] and Chen et al. [17]. More precisely, we let $\mathcal{M} = \{\mu \in \mathbb{R}^d : \|\mu\|_2 \le B\}$ and define $\mathcal{P}(\mu)$ to be the distribution of $(x, y)$, where $(x_0, y) \sim \mathcal{P}(0)$ and $x = x_0 + \mu$. Then, $R(\mu, \theta) = \mathbb{E}_{z \sim \mathcal{P}(\mu)} r(z; \theta) = -\mu^\top \theta$ and $L(\mu, \theta) = \mathbb{E}_{z \sim \mathcal{P}(\mu)} \ell(z; \theta)$.

We prove that both the decision-maker and the agents prefer the agents' equilibrium.

**Proposition 4.1.** *Assume the linear regression setup described above. Then, we have*

$$\frac{\sigma^2}{2} + \frac{\|\beta\|_2^2 \min(1, B)^2}{2(1 + \min(1, B)^2)} = L(\mu_{\mathrm{SE}}, \theta_{\mathrm{BR}}(\mu_{\mathrm{SE}})) \le \mathrm{SR}_L(\theta_{\mathrm{SE}}) = \frac{\sigma^2}{2} + \frac{\|\beta\|_2^2 B^2}{2(1 + B^2)},$$

$$-\frac{\|\beta\|_2 \min(1, B)}{1 + \min(1, B)^2} = \mathrm{SR}_R(\mu_{\mathrm{SE}}) \le R(\mu_{\mathrm{BR}}(\theta_{\mathrm{SE}}), \theta_{\mathrm{SE}}) = -\frac{\|\beta\|_2 B}{1 + B^2}.$$

When $B \le 1$, the losses implied by the two scenarios are the same, while when $B > 1$, having the agents lead is strictly better for both players. Moreover, the strategic agents' manipulation cost is no higher when they lead: $\|\mu_{\mathrm{SE}}\|_2 \le \|\mu_{\mathrm{BR}}(\theta_{\mathrm{SE}})\|_2$.

## 4.2 Logistic regression

Next we consider a classification example. Suppose that the non-strategic data $(x_0, y)$ is sampled according to a base joint distribution $\mathcal{P}(0)$ supported on $\mathbb{R}^d \times \{0, 1\}$. Unlike in the linear regression example, we place no further constraint on $\mathcal{P}(0)$.

We assume that the decision-maker trains a logistic regression classifier:

$$\ell(z; \theta) = -y x^\top \theta + \log(1 + e^{x^\top \theta}).$$

The agents with $y = 0$ can manipulate their features to increase the probability of being positively labeled. A similar setup is considered by Dong et al. [20]. As in the previous example, the agents have a limited budget to change their features: if their non-strategic features are $x_0$, they can move to any $x$ which is at distance at most $B$ from $x_0$, $\|x - x_0\|_2 \le B$. Thus, we set $\mathcal{M} = \{\mu \in \mathbb{R}^d : \|\mu\|_2 \le B\}$ and denote by $\mathcal{P}(\mu)$ the joint distribution of $(x, y)$ where $(x_0, y) \sim \mathcal{P}(0)$ and $x = x_0 + \mu \mathbf{1}\{y = 0\}$. We let $R(\mu, \theta) = -\mu^\top \theta$ and $L(\mu, \theta) = \mathbb{E}_{z \sim \mathcal{P}(\mu)} \ell(z; \theta)$.

**Proposition 4.2.** *Assume the logistic regression setup described above. Then, we have*

$$L(\mu_{\mathrm{SE}}, \theta_{\mathrm{BR}}(\mu_{\mathrm{SE}})) \le \mathrm{SR}_L(\theta_{\mathrm{SE}}) \ and \ \mathrm{SR}_R(\mu_{\mathrm{SE}}) \le R(\mu_{\mathrm{BR}}(\theta_{\mathrm{SE}}), \theta_{\mathrm{SE}}).$$

There exist configurations of parameters such that the inequalities in Proposition 4.2 are strict, meaning that both players strictly prefer the agents to lead. We illustrate this empirically. In Figure 1 we generate non-strategic data according to $y \sim \mathrm{Bern}(p)$ and $x_0|y \sim N(4y - 2, 1)$ and plot the difference in risk between the two equilibria for the decision-maker and the agents, for varying $B$ and $p$. For large $p$ and small $B$, we see no difference between the equilibria. However, as $p$ decreases and $B$ increases, it becomes suboptimal for both players if the decision-maker leads.

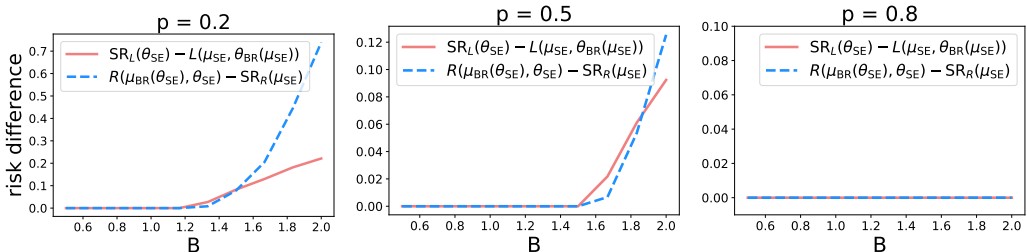

Figure 1: Difference in decision-maker's and agents' risk implied by the two Stackelberg equilibria, for different values of $B$ and $p$.

## 5 Discussion

We have shown how the consideration of update frequencies allows natural learning dynamics to converge to Stackelberg equilibria where either player can act as the leader. Moreover, we observed that the previously unexplored order of play in which the *agents lead* can result in lower risk for both players. We have only begun to understand the implications of reversing the order of play in strategic classification, and update frequencies in general, and many questions remain open for future work.

In social settings, there are many considerations and concerns beyond minimizing risk. While our preliminary observations suggest that reversing the order of play might have benefits, we have yet to fully understand the impact of this reversed order on the population interacting with the model. That said, we do not propose a new type of interaction; real-world decision-making algorithms already possess, and employ, the power to be reactive. Our new framework is simply flexible enough to capture the difference between proactive and reactive decision-makers.

Furthermore, we assume that the agents act on a fixed timescale. Sometimes it is possible for the agents to choose their timescale *strategically*. In that case, there is first a "meta-game" between the decision-maker and the agents, as they might compete for the leader/follower role. For example, if both prefer to lead, then both might aim to make slow updates to reach the leader position; perhaps surprisingly, this incentive might prevent any interaction at all.

Finally, we study order-of-play preferences only in linear/logistic regression with linear agent utilities. There are many other learning settings and classes of agents' utilities and costs in the literature, and going forward it is important to obtain general conditions when leading (or following) is preferable.

## Acknowledgements

We thank Moritz Hardt for an inspiring discussion and helpful feedback on this project. We wish to acknowledge support from the Office of Naval Research under the Vannevar Bush Fellowship program and support from HICON-LEARN (Design of HIgh CONfidence LEARNing-Enabled Systems), Defense Advanced Research Projects Agency award number FA8750-18-C-0101.

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
