# A   Experiments

As proof of concept, we demonstrate our theoretical findings empirically in a simulated logistic regression setting. In the first set of experiments, we adopt the model from Section 4.2 where agents are constrained in how they modify their features. In the second set of experiments we adopt a model more akin to that in [20] where the negatively classified agents are penalized from deviating from their true features. For simplicity, we refer to the first setting as the "constrained agent" case, and the second as the "costly deviations" case. The details of all numerical results are deferred to the end of this section.

## A.1   Agents with constraints

To begin, we verify our theoretical findings from Section 4.2. First we let the decision-maker lead and the agents follow, and then we switch the roles. In both cases the slower player runs the derivative-free update (3), and the faster player runs standard (projected) gradient descent. We generate 100 samples, fix $\alpha = 2$, $\sigma = 1$, $d = 1$, and vary $B$ and $p$. We run the interaction for a total of $T = 100000$ epochs, with each epoch of length $\tau = 200$. In Figure 2 and Figure 3 we plot the decision-maker's and the agents' average running risk against the number of epochs, for the two different orders of play, for $B = 2$ and $B = 1$, respectively. For $p \in \{0.1, 0.5\}$ and $B = 2$, we observe a clear gap between leading and following, the agents leading being the preferred order for both players. For $p = 0.9$ or $B = 1$, the two equilibria coincide asymptotically, however for any finite number of epochs both players still prefer the agents to lead.

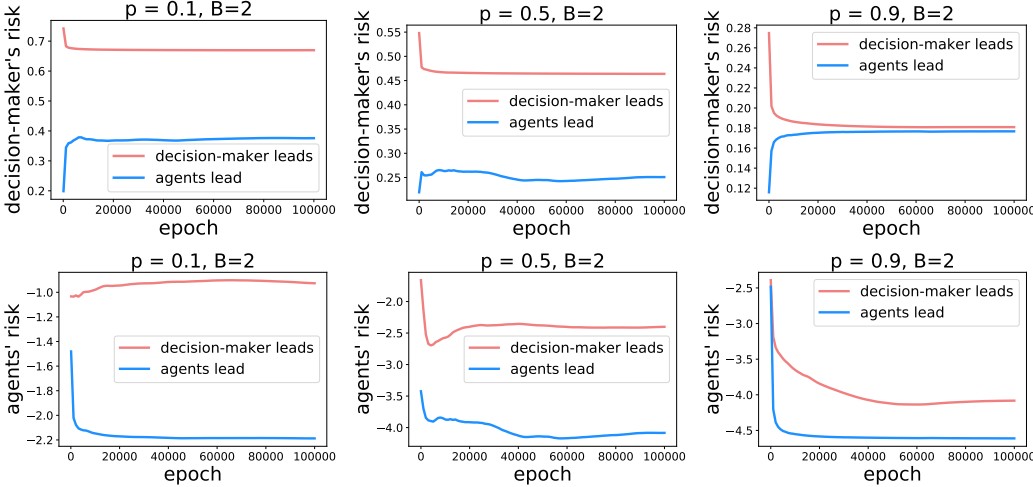

Figure 2: Decision-maker's and agents' average running risk for varying $p$ and $B = 2$.

## A.2   Agents with costly deviations

In this section, we verify our findings on a model where the decision-maker's problem is the same logistic regression problem posed in Section 4.2, but the strategic agents are penalized for deviating from their true features. In particular, the agents' risk $R$ takes the form:

$$R(\mu, \theta) = \frac{\lambda}{2}\|\mu\|^2 - \mu^T \theta.$$

We remark that though this setup is conceptually very similar to that in Section 4.2 (increasing $\lambda$'s can be seen as shrinking the constraint set), we use it to highlight that the experimental results are not caused by interactions with the constraints. Further, this setup is more readily comparable to previous models studied in, e.g., [20].

In our experiments we once again let the decision-maker lead and the agents follow, and then we switch the roles. In both cases the slower player runs the derivative-free update (3), and the faster player runs standard (projected) gradient descent. We generate 100 samples in $\mathbb{R}^2$, fix $\alpha = 1.5[1, 1]^T$

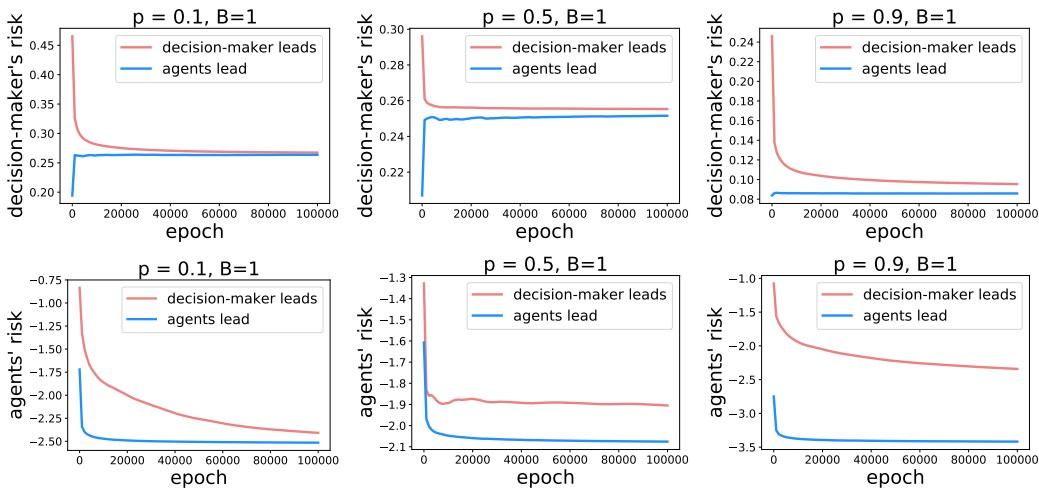

Figure 3: Decision-maker's and agents' average running risk for varying $p$ and $B = 1$.

and $\sigma = 1$, and vary $\lambda$ and $p$. We run the interaction for a total of $T = 50000$ epochs, with each epoch of length $\tau = 100$. In Figure 4 and Figure 5 we plot the decision-maker's and the agents' average running risk against the number of epochs, for the two different orders of play and for $\lambda = 1$ and $\lambda = 20$ respectively. To be able to analyze the long-run behavior, we also explicitly compute the Stackelberg risks of the decision-maker and strategic agents and find the global minima which correspond to the decision-maker's and agents' equilibria respectively.

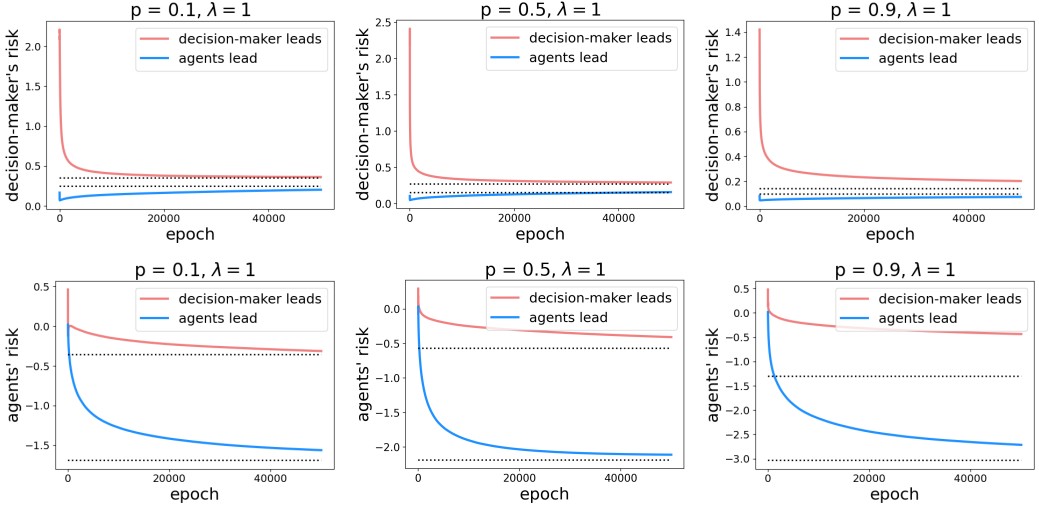

Figure 4: Decision-maker's and agents' average running risk for varying $p$ and $\lambda = 1$.

In Figure 4 we empirically observe that there is gap between the decision-maker's risk at their Stackelberg equilibrium and at the agents', and that the decision-maker consistently achieves a lower risk when the agents lead. Further, we note that agents consistently prefer leading, meaning that *both* the decision-maker and agents prefer if the order-of-play is flipped. We also observe that the proposed dynamics converge to the desired equilibria, validating our theoretical results.

**Remark A.1.** *Our empirical results suggest that the agents' equilibrium is a strictly better equilibrium in terms of the social cost (defined classically in game theory as the sum of the agents' and decision-maker's risks). Such equilibria were not captured by prior models and only emerge when considering learning dynamics of* both *players.*

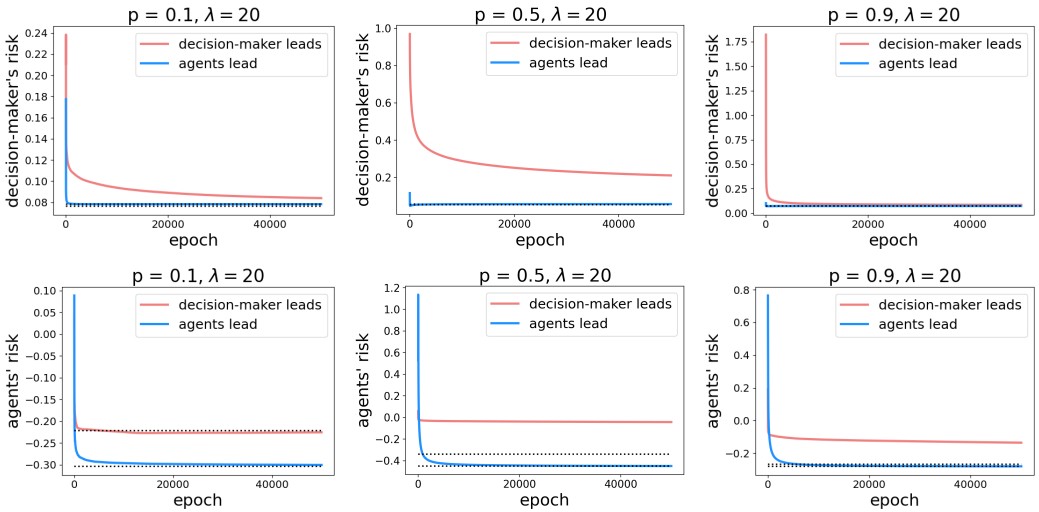

Figure 5: Decision-maker's and agents' average running risk for varying $p$ and $\lambda = 20$.

In Figure 5 we again observe that the proposed dynamics converge to the Stackelberg equilibria. By comparing with Figure 4 we also observe that as $\lambda$ is increased and $p$ increases, the gap between the two equilibria shrinks and disappears entirely when $p = 0.9$ and $\lambda = 20$. This is similar to the behavior seen in the constrained agent problem where shrinking the constraint set can give rise to Nash equilibria where neither agent strictly prefers leading or following. This highlights the inter-dependence between the preferred order-of-play and the problem structure, meaning that understanding when leading or following is strictly preferred by players is a non-trivial learning problem.

### A.3 Further experimental details

To generate Figure 1, we first find the decision-maker's equilibrium by optimizing $L\left(\frac{\theta}{\|\theta\|_2}B, \theta\right)$, as given in the proof of Proposition 4.2, with 1000 steps of gradient descent. We approximate the relevant expectation via a sample average over 1000 samples $x_0 \sim N(\alpha, \sigma^2)$. Once we have $\theta_{\text{SE}}$, we compute $\mu_{\text{BR}}(\theta_{\text{SE}}) = \frac{\theta_{\text{SE}}}{\|\theta_{\text{SE}}\|_2}B$. To find the agents' equilibrium, we perform grid search over 1000 equally spaced points in the interval $[-B, B]$. For each point in the grid, we compute the relevant best response $\theta_{\text{BR}}(\mu)$ by running 1000 steps of gradient descent, again estimating the expectation over 1000 samples. We take as the agents' equilibrium the point $\mu$ in the grid that minimizes the estimated value of $R(\mu, \theta_{\text{BR}}(\mu))$.

To generate Figures 2 and 3, in all experiments we set the step size of the faster player, that is, the one running gradient descent, to $0.1$. When the decision-maker leads, we set their step size to be $\eta_t = 0.5t^{-3/4}$. When the agents lead, we set their step size to be $\eta_t = 0.05t^{-3/4}$. We let the perturbation parameter $\delta$ decrease with time, and set $\delta_t = t^{-1/4}$.

To generate Figure 4, when the decision-maker led, we set their step size to be $\eta_t = 0.1t^{-3/4}$. When the agents led, we set their step size to be $\eta_t = 0.01t^{-3/4}$. We let the perturbation parameter $\delta$ decrease with time, and set $\delta_t = t^{-1/4}$. When the decision-maker and agents followed, their stepsizes for gradient descent were $0.1$ and $0.01$ respectively. To generate Figure 5 all parameters were the kept the same except when the decision-maker led, we set their step size to be $\eta_t = t^{-3/4}$.

To compute the decision-maker's and agents' risk at the decision-maker's equilibrium, we explicitly computed the decision-maker's Stackelberg risk by using the fact that the best response of the agents is $\mu_{\text{BR}}(\theta) = \frac{1}{\lambda}\theta$. We then minimized this risk directly using gradient descent with a fixed step-size of $0.1$.

To compute the decision-maker's and agents' risk at the agents' equilibrium, we computed their Stackelberg risk over a grid by fixing $\mu$ and running gradient descent with stepsize $0.1$ on the decision-

maker's problem until convergence for each value of $\mu$. We then found the minimum of the agents' Stackelberg risk by searching over the grid.

## B  Proofs

**Lemma B.1.** *Suppose that $\mathcal{M}$ is compact. If the decision-maker is proactive and the strategic agents' actions satisfy condition* (A1)*, then*

$$\lim_{\tau \to \infty} \frac{1}{\tau} \sum_{j=1}^{\tau} \mathbb{E}\|\mu_{j,\tau} - \mu_{\mathrm{BR}}(\theta_t)\|_2 = 0.$$

*Similarly, if the decision-maker is reactive and*

$$\lim_{T \to \infty} \lim_{\tau \to \infty} \frac{1}{T} \sum_{t=1}^{T} \mathbb{E}\mathrm{SR}_R(\mu_t) - \mathrm{SR}_R(\mu_{\mathrm{SE}}) = 0,$$

*then*

$$\lim_{T \to \infty} \lim_{\tau \to \infty} \frac{1}{T} \sum_{t=1}^{T} \mathbb{E}\|\mu_t - \mu_{\mathrm{SE}}\|_2 = 0.$$

*Proof.* We will prove the second statement; the proof of the first statement is completely analogous.

By the uniqueness of $\mu_{\mathrm{SE}}$ and compactness of $\mathcal{M}$, notice that for all $\mu$ and $\epsilon > 0$ such that $\|\mu - \mu_{\mathrm{SE}}\|_2 \geq \epsilon$, we have $\mathrm{SR}_R(\mu) - \mathrm{SR}_R(\mu_{\mathrm{SE}}) \geq \delta(\epsilon) > 0$, for some $\delta(\epsilon)$. We will use this observation to argue that, if $\frac{1}{T} \sum_{t=1}^{T} \mathbb{E}\|\mu_t - \mu_{\mathrm{SE}}\|_2 \not\to 0$, then that must imply positive regret in the limit, which concludes the proof by contradiction.

Denote $\mathrm{dist}_t = \lim_{\tau \to \infty} \mathbb{E}\|\mu_t - \mu_{\mathrm{SE}}\|_2$, and suppose that

$$\lim_{T \to \infty} \frac{1}{T} \sum_{t=1}^{T} \mathrm{dist}_t \neq 0.$$

Then, that implies that for every $\epsilon > 0$, there is a sequence $\{a_k\}_{k=1}^{\infty}$ such that $\frac{1}{a_k} \sum_{t=1}^{a_k} \mathrm{dist}_t > \epsilon$ for all $k$. Fix $0 < \epsilon' < \epsilon$, and denote $p_k = \frac{1}{a_k}|\{t \leq a_k : \mathrm{dist}_t > \epsilon'\}|$. Then, we have

$$\epsilon < \frac{1}{a_k} \sum_{t=1}^{a_k} \mathrm{dist}_t \leq p_k D_{\mathcal{M}} + \epsilon',$$

where $D_{\mathcal{M}} = \max_{\mu,\mu' \in \mathcal{M}} \|\mu - \mu'\|_2$. Therefore, $p_k \geq \frac{\epsilon - \epsilon'}{D_{\mathcal{M}}} > 0$. This shows that in the sum $\frac{1}{a_k} \sum_{t=1}^{a_k} \mathrm{dist}_t$ there is a *constant* fraction of terms outside a ball of radius $\epsilon'$ around $\mu_{\mathrm{SE}}$, in expectation. Fix one such term $\mathrm{dist}_{t^*}$. Then, we know

$$\epsilon' \leq \mathrm{dist}_{t^*} \leq \lim_{\tau \to \infty} \mathbb{P}\{\|\mu_{t^*} - \mu_{\mathrm{SE}}\|_2 \geq \epsilon'/2\} D_{\mathcal{M}} + \epsilon'/2.$$

Therefore, we can conclude that $\lim_{\tau \to \infty} \mathbb{P}\{\|\mu_{t^*} - \mu_{\mathrm{SE}}\|_2 \geq \epsilon'/2\} \geq \frac{\epsilon'}{2D_{\mathcal{M}}} > 0$. On this event, we also know that $\lim_{\tau \to \infty} \mathrm{SR}_R(\mu_{t^*}) - \mathrm{SR}_R(\mu_{\mathrm{SE}}) > \delta(\epsilon'/2)$. Putting everything together, we have shown that

$$\frac{1}{a_k} \sum_{t=1}^{a_k} \lim_{\tau \to \infty} \mathbb{E}\mathrm{SR}_R(\mu_t) - \mathrm{SR}_R(\mu_{\mathrm{SE}}) \geq \Delta > 0,$$

and this holds for all terms in the sequence $\{a_k\}$. This finally implies that $\frac{1}{T} \sum_{t=1}^{T} \mathbb{E}\mathrm{SR}_R(\mu_t) - \mathrm{SR}_R(\mu_{\mathrm{SE}}) \not\to 0$. Since this contradicts the hypothesis, we conclude that $\lim_{T \to \infty} \lim_{\tau \to \infty} \frac{1}{T} \sum_{t=1}^{T} \mathbb{E}\|\mu_t - \mu_{\mathrm{SE}}\|_2 = 0$. $\qquad\square$

## B.1 Proof of Theorem 3.1

We let $\widehat{\mathrm{SR}}_L(\theta) = \mathbb{E}_{v \sim \mathrm{Unif}(\mathcal{B})}[\mathrm{SR}_L(\theta + \delta v)]$, where $\mathcal{B}$ denotes the unit ball. Then, we know that

$$\nabla \widehat{\mathrm{SR}}_L(\theta) = \frac{d}{\delta} \mathbb{E}_{u \sim \mathcal{S}}[\mathrm{SR}_L(\theta + \delta u) u],$$

where $\mathcal{S}$ denotes the unit sphere. Denote by $\hat{\theta}_{\mathrm{SE}}$ the optimum of $\widehat{\mathrm{SR}}_L$, and notice that $\widehat{\mathrm{SR}}_L$ is convex since $\mathrm{SR}_L$ is convex.

For any fixed $t$, we have

$$\|\phi_{t+1} - \hat{\theta}_{\mathrm{SE}}\|_2^2 \leq \|\phi_t - \eta_t \frac{d}{\delta} L(\bar{\mu}_t, \phi_t + \delta u_t) u_t - \hat{\theta}_{\mathrm{SE}}\|_2^2$$

$$\leq \|\phi_t - \hat{\theta}_{\mathrm{SE}}\|_2^2 - 2\eta_t \frac{d}{\delta} L(\bar{\mu}_t, \phi_t + \delta u_t) u_t^\top (\phi_t - \hat{\theta}_{\mathrm{SE}}) + \eta_t^2 \frac{d^2}{\delta^2} \|L(\bar{\mu}_t, \phi_t + \delta u_t) u_t\|_2^2$$

$$\leq \|\phi_t - \hat{\theta}_{\mathrm{SE}}\|_2^2 - 2\eta_t \frac{d}{\delta} L(\bar{\mu}_t, \phi_t + \delta u_t) u_t^\top (\phi_t - \hat{\theta}_{\mathrm{SE}}) + \eta_t^2 \frac{d^2 B^2}{\delta^2}. \tag{6}$$

Focusing on the middle term, we have

$$L(\bar{\mu}_t, \phi_t + \delta u_t) u_t^\top (\phi_t - \hat{\theta}_{\mathrm{SE}}) = L(\bar{\mu}_t, \phi_t + \delta u_t) u_t^\top (\phi_t - \hat{\theta}_{\mathrm{SE}}) \pm L(\mu_{\mathrm{BR}}(\theta_t), \phi_t + \delta u_t) u_t^\top (\phi_t - \hat{\theta}_{\mathrm{SE}})$$

$$\geq L(\mu_{\mathrm{BR}}(\phi_t + \delta u_t), \phi_t + \delta u_t) u_t^\top (\phi_t - \hat{\theta}_{\mathrm{SE}}) - \beta_\mu \|\bar{\mu}_t - \mu_{\mathrm{BR}}(\theta_t)\|_2 D_\Theta.$$

Denote $\epsilon_t \overset{\text{def}}{=} \mathbb{E}\|\bar{\mu}_t - \mu_{\mathrm{BR}}(\theta_t)\|_2$. Taking expectations of both sides, we get

$$\mathbb{E} L(\bar{\mu}_t, \phi_t + \delta u_t) u_t^\top (\phi_t - \hat{\theta}_{\mathrm{SE}}) \geq L(\mu_{\mathrm{BR}}(\phi_t + \delta u_t), \phi_t + \delta u_t) u_t^\top (\phi_t - \hat{\theta}_{\mathrm{SE}}) - \beta_\mu D_\Theta \epsilon_t.$$

Going back to equation (6) and taking expectations of both sides, we get

$$\mathbb{E}\|\phi_{t+1} - \hat{\theta}_{\mathrm{SE}}\|_2^2 \leq \mathbb{E}\|\phi_t - \hat{\theta}_{\mathrm{SE}}\|_2^2 - 2\eta_t (\mathbb{E}[\nabla \widehat{\mathrm{SR}}_L(\phi_t)^\top (\phi_t - \hat{\theta}_{\mathrm{SE}})] - \beta_\mu D_\Theta \epsilon_t) + \eta_t^2 \frac{d^2 B^2}{\delta^2}$$

$$\leq \mathbb{E}\|\phi_t - \hat{\theta}_{\mathrm{SE}}\|_2^2 - 2\eta_t (\mathbb{E}\widehat{\mathrm{SR}}_L(\phi_t) - \widehat{\mathrm{SR}}_L(\hat{\theta}_{\mathrm{SE}}) - \beta_\mu D_\Theta \epsilon_t) + \eta_t^2 \frac{d^2 B^2}{\delta^2},$$

where in the last line we use the fact that $\widehat{\mathrm{SR}}_L$ is convex. After rearranging, we have

$$\mathbb{E}\widehat{\mathrm{SR}}_L(\phi_t) - \widehat{\mathrm{SR}}_L(\hat{\theta}_{\mathrm{SE}}) \leq \frac{1}{2\eta_t} \left( \mathbb{E}\|\phi_t - \hat{\theta}_{\mathrm{SE}}\|_2^2 - \mathbb{E}\|\phi_{t+1} - \hat{\theta}_{\mathrm{SE}}\|_2^2 \right) + \frac{\eta_t d^2 B^2}{2\delta^2} + \beta_\mu D_\Theta \epsilon_t.$$

Summing up over $t \in \{1, \ldots, T\}$, we get

$$\sum_{t=1}^T (\mathbb{E}[\widehat{\mathrm{SR}}_L(\phi_t)] - \widehat{\mathrm{SR}}_L(\hat{\theta}_{\mathrm{SE}})) \leq \frac{D_\Theta^2}{2\eta_1} + \frac{1}{2} \sum_{t=1}^{T-1} \left( \frac{1}{\eta_{t+1}} - \frac{1}{\eta_t} \right) D_\Theta^2 + \frac{d^2 B^2}{2\delta^2} \sum_{t=1}^T \eta_t + \beta_\mu D_\Theta \sum_{t=1}^T \epsilon_t$$

$$\leq \frac{D_\Theta^2}{2\eta_T} + \frac{d^2 B^2}{2\delta^2} \sum_{t=1}^T \eta_t + \beta_\mu D_\Theta \sum_{t=1}^T \epsilon_t,$$

where we use the fact that $\eta_t$ is non-increasing.

We use the fact that $\mathrm{SR}_L$ is Lipschitz to bound the difference between $\mathrm{SR}_L$ and $\widehat{\mathrm{SR}}_L$:

$$\left| \mathbb{E}[\widehat{\mathrm{SR}}_L(\phi_t) - \mathrm{SR}_L(\theta_t)] \right| \leq 2\beta\delta,$$

and similarly

$$\min_{\theta \in \Theta}(\widehat{\mathrm{SR}}_L(\theta) - \mathrm{SR}_L(\theta) + \mathrm{SR}_L(\theta)) \geq \min_\theta \mathrm{SR}_L(\theta) - \beta\delta.$$

Putting everything together, we conclude

$$\sum_{t=1}^T (\mathbb{E}[\mathrm{SR}_L(\theta_t)] - \mathrm{SR}_L(\theta_{\mathrm{SE}})) \leq \frac{D_\Theta^2}{2\eta_T} + \frac{d^2 B^2}{2\delta^2} \sum_{t=1}^T \eta_t + 3\beta\delta T + \beta_\mu D_\Theta \sum_{t=1}^T \epsilon_t.$$

Setting $\eta_t = \eta_0 d^{-\frac{1}{2}} t^{-\frac{3}{4}}$ and $\delta = \delta_0 d^{\frac{1}{2}} T^{-1/4}$ yields the final bound:

$$\sum_{t=1}^{T} (\mathbb{E}[\mathrm{SR}_L(\theta_t)] - \mathrm{SR}_L(\theta_{\mathrm{SE}})) \leq \left( \frac{D_\Theta^2}{2\eta_0} + \frac{2B^2}{\delta_0^2} \right) \sqrt{d} T^{3/4} + \beta_\mu D_\Theta \sum_{t=1}^{T} \epsilon_t.$$

For the second statement, observe that

$$\|\bar{\mu}_t - \mu_{\mathrm{BR}}(\theta_t)\|_2 \leq \frac{1}{\tau} \sum_{j=1}^{\tau} \|\mu_{t,j} - \mu_{\mathrm{BR}}(\theta)\|_2,$$

and the right-hand side tends to 0 in expectation as $\tau \to \infty$ by Lemma B.1.

## B.2 Proof of Theorem 3.4

By standard convergence guarantees of gradient descent on PL objectives [34], we have

$$\|\mu_t - \mu_{\mathrm{BR}}(\theta_t)\|_2 \leq \sqrt{\kappa}(1 - \gamma\eta_\mu)^{\tau/2} \|\mu_{t-1} - \mu_{\mathrm{BR}}(\theta_t)\|_2,$$

where $\kappa \overset{\text{def}}{=} \frac{\beta_\mu^R}{\gamma}$. Denote $\epsilon_t \overset{\text{def}}{=} \|\mu_t - \mu_{\mathrm{BR}}(\theta_t)\|_2$. We will show that $\epsilon_t$ decays fast enough due to the decay in $\eta_t$. In particular, we have

$$
\begin{aligned}
\epsilon_t = \|\mu_t - \mu_{\mathrm{BR}}(\theta_t)\|_2 &\leq \sqrt{\kappa}(1 - \gamma\eta_\mu)^{\tau/2} \|\mu_{t-1} - \mu_{\mathrm{BR}}(\theta_t)\|_2 \\
&= \sqrt{\kappa}(1 - \gamma\eta_\mu)^{\tau/2} \|\mu_{t-1} - \mu_{\mathrm{BR}}(\theta_{t-1}) + \mu_{\mathrm{BR}}(\theta_{t-1}) - \mu_{\mathrm{BR}}(\theta_t)\|_2 \\
&\leq \sqrt{\kappa}(1 - \gamma\eta_\mu)^{\tau/2} \left( \|\mu_{t-1} - \mu_{\mathrm{BR}}(\theta_{t-1})\|_2 + \|\mu_{\mathrm{BR}}(\theta_{t-1}) - \mu_{\mathrm{BR}}(\theta_t)\|_2 \right) \\
&\leq \sqrt{\kappa}(1 - \gamma\eta_\mu)^{\tau/2} \|\mu_{t-1} - \mu_{\mathrm{BR}}(\theta_{t-1})\|_2 \\
&\quad + \sqrt{\kappa}(1 - \gamma\eta_\mu)^{\tau/2} \frac{\eta_t d\beta_{\mathrm{BR}}}{\delta} \|L(\mu_t, \phi_t + \delta u_t) u_t\|_2 \\
&\leq \sqrt{\kappa}(1 - \gamma\eta_\mu)^{\tau/2} \epsilon_{t-1} + \sqrt{\kappa}(1 - \gamma\eta_\mu)^{\tau/2} \frac{\eta_t d\beta_{\mathrm{BR}}}{\delta} B.
\end{aligned}
$$

Now suppose $\tau$ is chosen such that $\tau > \frac{\log(\kappa)}{\log\left( \frac{1}{1-\gamma\eta_\mu} \right)}$. Then we have that $\alpha(\tau) \overset{\text{def}}{=} \sqrt{\kappa}(1-\gamma\eta_\mu)^{\tau/2} < 1$. (Note that as $\tau$ increases, $\alpha(\tau)$ can be driven to zero.) Altogether, we find that:

$$\epsilon_t \leq \alpha(\tau)\epsilon_{t-1} + \alpha(\tau)\eta_t \frac{d\beta_{\mathrm{BR}} B}{\delta}.$$

Unrolling the recursion, we find that

$$\epsilon_t \leq \alpha(\tau)^t \epsilon_0 + \frac{d\beta_{\mathrm{BR}} B}{\delta} \sum_{i=1}^{t} \alpha(\tau)^{t+1-i} \eta_i.$$

Summing up over $t \in \{1, \dots, T\}$, we get

$$
\begin{aligned}
\sum_{t=1}^{T} \epsilon_t &\leq \epsilon_0 \sum_{t=1}^{T} \alpha(\tau)^t + \frac{d\beta_{\mathrm{BR}} B}{\delta} \sum_{t=1}^{T} \sum_{i=1}^{t} \alpha(\tau)^{t+1-i} \eta_i \\
&\leq \frac{\epsilon_0}{1 - \alpha(\tau)} + \frac{d\beta_{\mathrm{BR}} B}{\delta} \sum_{t=1}^{T} \sum_{i=1}^{T} \alpha(\tau)^{t+1-i} \eta_i \mathbf{1}\{i \leq t\} \\
&= \frac{\epsilon_0}{1 - \alpha(\tau)} + \frac{d\beta_{\mathrm{BR}} B}{\delta} \sum_{i=1}^{T} \eta_i \sum_{t=1}^{T} \alpha(\tau)^{t+1-i} \mathbf{1}\{i \leq t\} \\
&= \frac{\epsilon_0}{1 - \alpha(\tau)} + \frac{d\beta_{\mathrm{BR}} B}{\delta} \sum_{i=1}^{T} \eta_i \sum_{t=i}^{T} \alpha(\tau)^{t+1-i} \\
&\leq \frac{\epsilon_0}{1 - \alpha(\tau)} + \frac{d\beta_{\mathrm{BR}} B}{\delta(1 - \alpha(\tau))} \sum_{t=1}^{T} \eta_t.
\end{aligned}
$$

For $\eta_t = \eta_0 d^{-1/2} t^{-3/4}$ and $\delta = \delta_0 d^{1/2} T^{-1/4}$, we have

$$\sum_{t=1}^{T} \epsilon_t \leq \frac{1}{1 - \alpha(\tau)} \left( \epsilon_0 + \frac{4\beta_{\mathrm{BR}} B \eta_0 \sqrt{T}}{\delta_0} \right).$$

## B.3  Proof of Theorem 3.6

Define $\mu_t^* = D_t(\mu_1, \theta_{\mathrm{BR}}(\mu_1), \dots, \mu_{t-1}^*, \theta_{\mathrm{BR}}(\mu_{t-1}), \xi_t)$. First we will prove that

$$\lim_{T \to \infty} \lim_{\tau \to \infty} \frac{1}{T} \sum_{t=1}^{T} \mathbb{E} \mathrm{SR}_R(\mu_t) - \mathrm{SR}_R(\mu_{\mathrm{SE}}) = 0. \tag{7}$$

To show this, it suffices to prove that for all $t$, $\mu_t \to_p \mu_t^*$ as $\tau \to \infty$. The sufficiency of this condition follows because

$$\lim_{T \to \infty} \lim_{\tau \to \infty} \frac{1}{T} \sum_{t=1}^{T} \mathbb{E} \mathrm{SR}_R(\mu_t) - \mathrm{SR}_R(\mu_{\mathrm{SE}})$$

$$= \lim_{T \to \infty} \lim_{\tau \to \infty} \frac{1}{T} \sum_{t=1}^{T} [\mathbb{E} \mathrm{SR}_R(\mu_t) - \mathbb{E} \mathrm{SR}_R(\mu_t^*) + \mathbb{E} \mathrm{SR}_R(\mu_t^*)] - \mathrm{SR}_R(\mu_{\mathrm{SE}})$$

$$= \lim_{T \to \infty} \lim_{\tau \to \infty} \frac{1}{T} \sum_{t=1}^{T} (\mathbb{E} \mathrm{SR}_R(\mu_t) - \mathbb{E} \mathrm{SR}_R(\mu_t^*)),$$

where the last step follows by the assumption that the agents play a rational strategy. Therefore, if $\mu_t \to_p \mu_t^*$, continuity of $\mathrm{SR}_R(\mu)$ implies $\mathbb{E} \mathrm{SR}_R(\mu_t) - \mathbb{E} \mathrm{SR}_R(\mu_t^*) \to 0$ and we get the desired conclusion.

We prove that $\mu_t \to_p \mu_t^*$ by induction. Notice that $\mu_1 \equiv \mu_1^*$ by definition.

Suppose that $\mu_j \to_p \mu_j^*$ for all $j < t$. Denote by $\theta_{j,\tau}$ the possibly randomized algorithm that maps $\mu_j$ to $\theta_j$. Then, for any $\mu \in \mathcal{M}$, we know that $\|\theta_{j,\tau}(\mu) - \theta_{\mathrm{BR}}(\mu)\|_2 \to_p 0$ by assumption. This in turn implies that for all $j < t$,

$$\|\theta_{j,\tau}(\mu_j) - \theta_{\mathrm{BR}}(\mu_j^*)\|_2 \leq \|\theta_{j,\tau}(\mu_j) - \theta_{\mathrm{BR}}(\mu_j)\|_2 + \|\theta_{\mathrm{BR}}(\mu_j) - \theta_{\mathrm{BR}}(\mu_j^*)\|_2 \to_p 0,$$

where the second term tends to $0$ by the continuous mapping theorem. Finally, we can apply the continuity of $D_t$ to conclude that $\mu_t \to_p \mu_t^*$, as desired.

Let $\beta$ denote the Lipschitz constant of $\mathrm{SR}_L$. Finally, we we can apply this Lipschitz condition to conclude:

$$\frac{1}{T} \sum_{t=1}^{T} \mathbb{E} L(\mu_t, \theta_{t,\tau}) - L(\mu_{\mathrm{SE}}, \theta_{\mathrm{BR}}(\mu_{\mathrm{SE}}))$$

$$= \frac{1}{T} \sum_{t=1}^{T} [\mathbb{E} L(\mu_t, \theta_{t,\tau}) \pm \mathbb{E} L(\mu_t, \theta_{\mathrm{BR}}(\mu_t)))] - L(\mu_{\mathrm{SE}}, \theta_{\mathrm{BR}}(\mu_{\mathrm{SE}}))$$

$$= \frac{1}{T} \sum_{t=1}^{T} (\mathbb{E} L(\mu_t, \theta_{\mathrm{BR}}(\mu_t)) - L(\mu_{\mathrm{SE}}, \theta_{\mathrm{BR}}(\mu_{\mathrm{SE}})) + \mathbb{E}[L(\mu_t, \theta_{t,\tau}) - L(\mu_t, \theta_{\mathrm{BR}}(\mu_t))])$$

$$\leq \frac{\beta}{T} \sum_{t=1}^{T} \mathbb{E} \|\mu_t - \mu_{\mathrm{SE}}\|_2 + \frac{1}{T} \sum_{t=1}^{T} \mathbb{E}[L(\mu_t, \theta_{t,\tau}) - L(\mu_t, \theta_{\mathrm{BR}}(\mu_t))].$$

By Lemma B.1, the guarantee (7) implies that the first term vanishes. The second term vanishes by continuity. Therefore, taking the limit over $T, \tau$, we obtain

$$\lim_{T \to \infty} \lim_{\tau \to \infty} \frac{1}{T} \sum_{t=1}^{T} \mathbb{E} L(\mu_t, \theta_t) - L(\mu_{\mathrm{SE}}, \theta_{\mathrm{BR}}(\mu_{\mathrm{SE}})) = 0,$$

as desired.

## B.4 Proof of Propositon 4.1

First we assume the decision-maker leads. When $\theta$ is the deployed model, the best response by the agents is to simply move by distance $B$ in the direction of $\theta$. Thus, $\mu_{\mathrm{BR}}(\theta)$ is given by:

$$\mu_{\mathrm{BR}}(\theta) = \arg\min_{\mu} \mathbb{E}_{(x,y)\sim\mathcal{P}(\mu)} - x^\top\theta = \frac{\theta}{\|\theta\|_2} B.$$

This implies the following expected loss for the decision-maker:

$$L(\mu_{\mathrm{BR}}(\theta), \theta) = \mathbb{E}_{z\sim\mathcal{P}\left(\frac{\theta}{\|\theta\|_2}B\right)} \ell(z;\theta) = \frac{1}{2}\mathbb{E}_{(x_0,y)\sim\mathcal{P}(0)} \left(y - x_0^\top\theta - \|\theta\|_2 B\right)^2$$

$$= \frac{\sigma^2}{2} + \frac{1}{2}\|\beta - \theta\|_2^2 + \frac{B^2}{2}\|\theta\|_2^2.$$

This objective is convex and thus by finding a stationary point we observe that it is minimized at $\theta_{\mathrm{SE}} = \frac{\beta}{1+B^2}$. By plugging this choice back into the previous equation, we observe that the minimal Stackelberg risk of the decision-maker is equal to

$$\mathrm{SR}_L(\theta_{\mathrm{SE}}) = L(\mu_{\mathrm{BR}}(\theta_{\mathrm{SE}}), \theta_{\mathrm{SE}}) = \frac{\sigma^2}{2} + \frac{\|\beta\|_2^2 B^2}{2(1+B^2)}. \tag{8}$$

Moreover, the agents' loss at $\theta_{\mathrm{SE}}$ is equal to:

$$R(\mu_{\mathrm{BR}}(\theta_{\mathrm{SE}}), \theta_{\mathrm{SE}}) = -\|\theta_{\mathrm{SE}}\|_2 B = -\frac{\|\beta\|_2 B}{1+B^2}.$$

Now we reverse the order of play and assume that the agents lead. If the agents move by $\mu$, i.e. they follow the law $\mathcal{P}(\mu)$, then the decision-maker incurs loss:

$$L(\mu, \theta) = \mathbb{E}_{(x,y)\sim\mathcal{P}(\mu)}\frac{1}{2}\left(y - x_0^\top\theta - \mu^\top\theta\right)^2 = \frac{\sigma^2}{2} + \frac{1}{2}\|\beta - \theta\|_2^2 + \frac{1}{2}(\mu^\top\theta)^2.$$

By computing a stationary point, we find that the best response of the decision-maker is:

$$\theta_{\mathrm{BR}}(\mu) = (I + \mu\mu^\top)^{-1}\beta = \left(I - \frac{\mu\mu^\top}{1 + \|\mu\|_2^2}\right)\beta.$$

The Stackelberg risk of the strategic agent is then

$$\mathrm{SR}_R(\mu) = R(\mu, \theta_{\mathrm{BR}}(\mu)) = -\mu^\top\theta_{\mathrm{BR}}(\mu) = -\mu^\top\left(I - \frac{\mu\mu^\top}{1 + \|\mu\|_2^2}\right)\beta$$

$$= -\mu^\top\beta + \frac{\|\mu\|_2^2\mu^\top\beta}{1 + \|\mu\|_2^2} = -\frac{\mu^\top\beta}{1 + \|\mu\|_2^2}.$$

Among all $\mu$ such that $\|\mu\|_2 = C$, $\mathrm{SR}_R(\mu)$ is minimized when $\mu$ points in the $\beta$ direction: $\mu = C\frac{\beta}{\|\beta\|_2}$. With this reparameterization, we can equivalently write $\min_\mu \mathrm{SR}_R(\mu)$ as

$$\min_{C>0} \|\beta\|_2 \frac{-C}{1+C^2}.$$

This function is decreasing for $C \in (0,1]$, and increasing for $C > 1$. Therefore, $\mu_{\mathrm{SE}} = \min(1, B)\frac{\beta}{\|\beta\|_2}$, and

$$\mathrm{SR}_R(\mu_{\mathrm{SE}}) = -\|\beta\|_2\frac{\min(1, B)}{1 + \min(1, B)^2}.$$

Finally, we evaluate the decision-maker's loss at $\mu_{\mathrm{SE}}$:

$$L(\mu_{\mathrm{SE}}, \theta_{\mathrm{BR}}(\mu_{\mathrm{SE}})) = \frac{\sigma^2}{2} + \frac{1}{2}\frac{(\beta^\top\mu_{\mathrm{SE}})^2\|\mu_{\mathrm{SE}}\|_2^2}{(1 + \|\mu_{\mathrm{SE}}\|_2^2)^2} + \frac{1}{2}\left(\|\beta\|_2\frac{\min(1, B)}{1 + \min(1, B)^2}\right)^2$$

$$= \frac{\sigma^2}{2} + \frac{1}{2}\frac{\|\beta\|_2^2\min(1, B)^4}{(1 + \min(1, B)^2)^2} + \frac{1}{2}\left(\|\beta\|_2\frac{\min(1, B)}{1 + \min(1, B)^2}\right)^2$$

$$= \frac{\sigma^2}{2} + \frac{\|\beta\|_2^2\min(1, B)^2}{2(1 + \min(1, B)^2)}.$$

## B.5 Proof of Proposition 4.2

First we evaluate $L(\mu, \theta)$:

$$
\begin{aligned}
L(\mu, \theta) &= \mathbb{E}_{(x,y)\sim\mathcal{P}(\mu)}\left[-yx^\top\theta + \log(1 + e^{x^\top\theta})\right] \\
&= \mathbb{E}_{(x,y)\sim\mathcal{P}(\mu)}\left[\log(e^{-yx^\top\theta} + e^{(1-y)x^\top\theta})\right] \\
&= \mathbb{E}_{(x_0,y)\sim\mathcal{P}(0)}[\mathbf{1}\{y=1\}\log(1 + e^{-x_0^\top\theta}) + \mathbf{1}\{y=0\}\log(1 + e^{x_0^\top\theta+\mu^\top\theta})].
\end{aligned}
$$

We prove that the agents are never worse off if they lead. We will provide a sufficient condition; namely, we will show that

$$
\mathrm{SR}_R\left(\frac{\theta_{\mathrm{SE}}}{\|\theta_{\mathrm{SE}}\|_2}B\right) = R(\mu_{\mathrm{BR}}(\theta_{\mathrm{SE}}), \theta_{\mathrm{SE}}).
$$

This immediately implies that $\mathrm{SR}_R(\mu_{\mathrm{SE}}) \leq R(\mu_{\mathrm{BR}}(\theta_{\mathrm{SE}}), \theta_{\mathrm{SE}})$.

To see this, first observe that

$$
R(\mu_{\mathrm{BR}}(\theta_{\mathrm{SE}}), \theta_{\mathrm{SE}}) = B\|\theta_{\mathrm{SE}}\|_2,
$$

where $\theta_{\mathrm{SE}} = \arg\min_\theta L\left(\frac{\theta}{\|\theta\|_2}B, \theta\right)$. Here we use the fact that the best response of the agents is to simply move by distance $B$ in the direction of $\theta$:

$$
\mu_{\mathrm{BR}}(\theta) = \arg\max_{\mu\in\mathcal{M}}\theta^\top\mu = \frac{\theta}{\|\theta\|_2}B.
$$

By the fact that $\theta_{\mathrm{SE}}$ is a Stackelberg equilibrium, we know that $\nabla_\theta\mathrm{SR}_L(\theta_{\mathrm{SE}}) = 0$, where $\mathrm{SR}_L(\theta) = L(\frac{\theta}{\|\theta\|_2}B, \theta)$:

$$
\nabla_\theta\mathrm{SR}_L(\theta) = \mathbb{E}_{(x_0,y)\sim\mathcal{P}(0)}\left[\mathbf{1}\{y=1\}\frac{e^{-x_0^\top\theta}(-x_0)}{1 + e^{-x_0^\top\theta}} + \mathbf{1}\{y=0\}\frac{e^{x_0^\top\theta+\|\theta\|_2B}(x_0 + \frac{\theta}{\|\theta\|_2}B)}{1 + e^{x_0^\top\theta+\|\theta\|_2B}}\right].
$$

In contrast, consider $\nabla_\theta L(\mu, \theta)$:

$$
\nabla_\theta L(\mu, \theta) = \mathbb{E}_{(x_0,y)\sim\mathcal{P}(0)}\left[\mathbf{1}\{y=1\}\frac{e^{-x_0^\top\theta}(-x_0)}{1 + e^{-x_0^\top\theta}} + \mathbf{1}\{y=0\}\frac{e^{x_0^\top\theta+\mu^\top\theta}(x_0 + \mu)}{1 + e^{x_0^\top\theta+\mu^\top\theta}}\right].
$$

Notice that $\nabla_\theta\mathrm{SR}_L(\theta_{\mathrm{SE}}) = 0$ implies that $\nabla_\theta L(\frac{\theta_{\mathrm{SE}}}{\|\theta_{\mathrm{SE}}\|_2}B, \theta_{\mathrm{SE}}) = 0$. Since $L(\mu, \theta)$ is convex in $\theta$, this condition implies that $\theta_{\mathrm{SE}}$ is a best response to $\frac{\theta_{\mathrm{SE}}}{\|\theta_{\mathrm{SE}}\|_2}B$, hence

$$
\mathrm{SR}_R\left(\frac{\theta_{\mathrm{SE}}}{\|\theta_{\mathrm{SE}}\|_2}B\right) = R(\mu_{\mathrm{BR}}(\theta_{\mathrm{SE}}), \theta_{\mathrm{SE}}).
$$

Now we analyze the decision-maker's preference. Notice that $L(\mu, \theta)$ is increasing in $\mu^\top\theta$; that is, for any $\theta$ it holds that $\max_\mu L(\mu, \theta) = L(\mu_{\mathrm{BR}}(\theta), \theta)$. Using this, we observe that for every $\theta$ we have

$$
L(\mu_{\mathrm{SE}}, \theta_{\mathrm{BR}}(\mu_{\mathrm{SE}})) \leq L(\mu_{\mathrm{SE}}, \theta) \leq \max_{\mu\in\mathcal{M}} L(\mu, \theta) = \mathrm{SR}_L(\theta).
$$

Since this also holds for $\theta = \theta_{\mathrm{SE}}$, we conclude that following is never worse than leading for the decision-maker.

## C  Local guarantees when $\mathrm{SR}_L$ is nonconvex

We provide local guarantees in the limit when the decision-maker's Stackelberg risk is possibly nonconvex. Specifically, we show that the update rule (3) converges to a stationary point of a smooth version of the decision-maker's Stackelberg risk, provided that the strategic agents achieve iterate convergence. Moreover, under mild regularity conditions, this stationary point is a local minimum (see, e.g., Theorem 9.1 in [7]).

**Proposition C.1.** *Assume that the agents achieve iterate convergence, $\|\mu_t - \mu_{\mathrm{BR}}(\theta_t)\| \to 0$ almost surely as $t \to \infty$. Further, assume that the decision-maker's Stackelberg risk is Lipschitz and smooth, and that $L(\mu, \theta)$ is Lipschitz in its first argument and bounded for all $\mu$ and $\theta$. If the decision-maker runs update* (3) *with $\eta_t$ satisfying $\sum_{t=1}^{\infty} \eta_t = \infty$ and $\sum_{t=1}^{\infty} \eta_t^2 < \infty$, and $\delta = \frac{\epsilon}{4\beta}$, then as $t \to \infty$, $\phi_t \to \phi^*$ such that $\nabla \widehat{\mathrm{SR}}_L(\phi^*) = 0$, where $\widehat{\mathrm{SR}}_L(\phi) = \mathbb{E}_{v \sim \mathrm{Unif}(\mathcal{B})} [\mathrm{SR}_L(\phi + \delta v)]$.*

*Proof.* We make use of results from the literature on stochastic approximations (see, e.g., [10]).

As in the proof of Theorem 3.1, let $\widehat{\mathrm{SR}}_L(\phi) = \mathbb{E}_{v \sim \mathrm{Unif}(\mathcal{B})} [\mathrm{SR}_L(\phi + \delta v)]$, and recall $\theta_t = \phi_t + \delta u_t$, $u_t \sim \mathrm{Unif}(\mathcal{S}^{d-1})$.

We begin by writing the update for $\phi_t$ as:

$$
\phi_{t+1} = \phi_t - \eta_t \left( \nabla_\phi \widehat{\mathrm{SR}}_L(\phi_t) - \left( \underbrace{\nabla_\phi \widehat{\mathrm{SR}}_L(\phi_t) - \frac{d}{\delta} L(\mu_{\mathrm{BR}}(\theta_t), \theta_t) u_t}_{=\mathrm{I}} \right) \right)
$$
$$
- \eta_t \frac{d}{\delta} \left( \underbrace{L(\mu_t, \theta_t) u_t - L(\mu_{\mathrm{BR}}(\theta_t), \theta_t) u_t}_{=\mathrm{II}} \right)
$$

Since

$$
\nabla_\phi \widehat{\mathrm{SR}}_L(\phi) = \mathbb{E}_{u \sim \mathrm{Unif}(\mathcal{S}^{d-1})} \left[ \frac{d}{\delta} \mathrm{SR}_L(\phi + \delta u) u \right],
$$

we know $\mathbb{E}_u[\mathrm{I}] = 0$. Since $L$ is bounded and $\mathrm{SR}_L$ is smooth, we know $\|\mathrm{I}\|_2$ is bounded. Thus, I is a zero-mean random variable with finite variance.

For term II, we use the assumed Lipshchitzness of $L$ in the first argument to find that:

$$
\|\mathrm{II}\|_2 \leq \beta_\mu \|\mu_t - \mu_{\mathrm{BR}}(\theta_t)\|_2 \|u_t\|_2
$$
$$
= \beta_\mu \|\mu_t - \mu_{\mathrm{BR}}(\theta_t)\|_2,
$$

where we use the fact that $\|u\|_2 = 1$. By assumption, $\|\mu_t - \mu_{\mathrm{BR}}(\theta_t)\|_2 \to 0$ almost surely as $t \to \infty$. Thus, we can write the update rule as:

$$
\phi_{t+1} = \phi_t - \eta_t \left( \nabla_\phi \widehat{\mathrm{SR}}_L(\phi_t) + \xi_t + M_t \right),
$$

where $\xi_t = o(1)$ and $M_t$ is a zero-mean random variable with finite variance. Since the assumed choice of $\eta_t$ satisfies $\sum_{t=1}^{\infty} \eta_t = \infty$ and $\sum_{t=1}^{\infty} \eta_t^2 < \infty$ we can invoke Chapter 2, Corollary 3 in [10] to find that $\phi_t \to \phi^* \in \{\phi : \nabla_\phi \widehat{\mathrm{SR}}_L(\phi) = 0\}$. $\square$