# OpenReview forum: "Who Leads and Who Follows in Strategic Classification?"
_NeurIPS.cc/2021/Conference — NeurIPS 2021 Poster_

### Official Review · Reviewer_1kNE · 2021-07-05

**Rating:** 6
**Confidence:** 3

**Summary:**

The authors study an extension of strategic classification in which they consider how the time it takes the model designer and agents to react to one another effects the dynamics of strategic classification. Differences in the updating frequencies of the two parties can essentially switch which party is the leader (typically the model designer is the leader).  Different equilibria can arise depending on which party has the faster update frequency. Convergence to these equilibria is characterized in the case of agents leading and also the model designer leading. Additionally, cases in which having the agents be the leaders is desirable for both parties are outlined.

**Main Review:**

Strengths:

— The paper examines an important extension of the traditional strategic classification model. Namely consideration of response times, for both the model designer and the agents, adds a number of interesting insights.

— Significant contributions are made in the from of results pertaining to this model extension. Specifically, the authors quantify convergence of strategies for the case of both a reactive and proactive model designer, as well as providing examples in which the latter is preferable to both parties.

— The authors’ assumptions allow their results to be applicable in a wide range of settings.

— It is both counter intuitive and interesting that situations can arise in which both the model designer and the agents prefer the agents to lead. Aside from being generally interesting, this result implies that the strand strategic classification model (model designer leading) can lead to less desirable outcomes and thus motivates the need to consider reactive model design and potential tool for robustness.

— The paper is well written, the contributions are clearly outlined, and helpful intuition regarding results is provided.


Weaknesses:

— The informal presentation of Theorem 1.2 seems to overstate the actual result (which I believe is Proposition 4.1 and 4.2, although there appears to be no explicit mention of the formal version of 1.2), in that the theorem insinuates both parties prefer agents to lead when linear or logistic regression models are deployed. However, Section 4 appears to only provide sufficient conditions on the distribution of true features for this result to hold, rather than showing that it holds In general.

— When looking at agents as potentially spending large periods of time to manipulate their features, it could be the case that this process also manipulates their true label, e.g. spending a year doing actions to increase one’s credit score may inadvertently increase one’s true credit-worthiness. Without accounting for this potential shift in labels, the the utility of the model designer may be incorrectly modeled.

— Additionally if the update frequencies are low enough, such as some type of annual application system, we may be more interested in considering utility within some small number of iterations, rather than quantifying convergence as the number of iterations approach infinity. This is somewhat handled with several of the inequalities presented in section 3, but having a better characterization of utility with respect to each iteration would be helpful since the setting this paper studies is motivated by slow reacting agents.

**Time Spent Reviewing:**

4

---

> ### Author Response · Authors · 2021-08-10
> **Response to Reviewer 1kNE**
>
> Thank you for your comments about the presentation and our model. Below we address the weaknesses pointed out in the review:
>
> (Informal statement of Theorem 1.2)
>
> Indeed, Theorem 1.2 is meant to capture Propositions 4.1 and 4.2. While our intention was for this statement to be a very informal approximation of the true result, we agree that the current phrasing is too broad---thank you for bringing this to our attention. We will modify the statement to describe more accurately the assumptions on the strategic manipulations and agents’ data that underlie the result.
>
> (Modifying the true label)
>
> Our general theoretical results do not assume or imply that the agents’ labels cannot change; they are agnostic to changes in features vs labels (since both could be incorporated into $\mu$, the agents’ actions). It is only our illustrative examples in Section 4 that assume static labels, but these examples only serve to motivate the broader study of update frequencies and order of play in strategic classification.
>
> (Slow reacting agents)
>
> Our motivation in this paper is to analyze situations that emerge as online platforms begin deploying decision-making algorithms. As such, we are less interested in modeling systems that are updated annually, as suggested, but rather by online platforms that get updated every couple of hours or days. This is why we expect to see many epochs in a reasonably small time window. While we agree that finite-time bounds are more interesting, some of our results are asymptotic mainly because we make weak assumptions on agent behavior (i.e., no-regret learning). Providing additional non-asymptotic results would require placing stronger assumptions on the way agents learn.
>
> Since rephrasing Theorem 1.2 is a fairly minor edit, and the second and third stated weaknesses seem to have been simple misunderstandings that we will happily clarify, we hope that you will reevaluate our paper with these points in mind.

---

> ### Author Response · Authors · 2021-09-01
> **Feedback**
>
> Thanks for your hard work reviewing. We just wanted to reach out and see if you had any comments or questions after our response given that the discussion period is ending. We hope that we were able to resolve your concerns.

---

### Official Review · Reviewer_6sCC · 2021-07-07

**Rating:** 7
**Confidence:** 4

**Summary:**

Almost all of the literature on strategic classification is situated in a Stackelberg game setting in which the decision-maker leads, committing to a model, after which agents follow, best-responding to the model. The core insight of this work is that in some real life situations, the decision-maker is able to adjust their model more rapidly than agents are able to adjust their strategic adaptation, leading to a reversal of the traditional order of play: the agents become the leader, and the decision-maker becomes the follower. The authors compare the dynamics of these "proactive" and "reactive" regimes, which differ in the comparative frequency of the two parties' updates, under the weak assumption that agents are no-regret learners. In particular they give conditions under which the learning dynamics converge to Stackelberg equilibria. Finally, they show that in two simple examples (linear regression and logistic regression with bounded actions of linear cost), the equilibrium in which the agents lead is at least as preferable (and sometimes more preferable) for _both_ agents.

**Limitations And Societal Impact:**

One unaddressed limitation of the work is the assumption that agents act as a unit, maximizing their aggregate utility as a group. I think this is an understandable assumption for reasons of tractability and space. Indeed, when the decision-maker moves first, there is no difference between agents acting in their own self-interest versus in their collective self-interest, since each agent's utility is unaffected by the actions of others. But when the agents move first, the action of one agent can affect the model deployed by the decision-maker and thus the utility of another agent, and so the assumption matters and is in my opinion unrealistic (hence the common use of perfect Bayesian equilibria in the signaling literature).

**Main Review:**

This paper makes several contributions:
1) Pointing out that the order of play in strategic classification can be reversed
2) ...and that differences of frequency of updates can determine the order of play
3) Showing that both players might prefer the order in which the agents lead
4) Performing analyses that merely assume agents achieve vanishing regret in comparison to best-response, rather than the traditional stronger assumption that agents instantly play the optimal move

A very relevant recent paper seems to have been missed by the authors: ["Scoring Strategic Agents"](https://arxiv.org/abs/1909.01888) by Ian Ball. In this work, Ball studies strategic regression under both orders of play: decision-maker first, and agents first. These correspond to the classic economic settings of _screening_ and _signaling_ respectively. Thus, the current paper is not in fact the first to vary the order of play in the literature on strategic classification, so contribution #1 is not wholly novel.

That notwithstanding, I am recommending acceptance because contributions #2,3,4 are novel as far as I know, and are substantive and interesting. #2 and #3 are neat observations that are presented convincingly. The authors make a good case that the agents-first model is a better fit for many real-life settings than the decision-maker-first model. (The main body of the paper as a whole is admirably well-written). Regarding #4, I appreciate the no-regret model of agent behavior; I would like to see more strategic classification works use this rather than assume agents can instantly best-respond. Moreover, order of play is acutely underdiscussed in the literature even if it has already been brought up once before, and the Ball paper is framed as a work of economics rather than computer science.

One thing that confused me: the authors use a parameter $\mu$ to denote "the agents' action." But of course in many cases different agents take different actions. In the logistic regression example, $\mu$ refers to the one action all the true-negative agents take (the true-positives don't take any action). So I gather that $\mu$ is meant to parameterize the actions collectively taken by the full distribution of agents? It would be nice to have clarity about this from the authors.

I have only given the proofs a cursory skim, not checked them for correctness.

**Time Spent Reviewing:**

7

---

> ### Author Response · Authors · 2021-08-10
> **Response to Reviewer 6sCC**
>
> Thank you for taking the time to review our paper and for your comments. Hopefully we can clear up your lingering concerns.
>
> Thank you for the reference to the paper by Ian Ball; we were unfamiliar with the work, but it is indeed related to ours and we will cite it accordingly. The model analyzed seems different than that normally addressed in strategic classification---the role of the intermediary is particularly interesting---but the analysis of the signaling and screening settings is really quite compelling. While the paper analyzes a one-shot game and not the case where agents must learn through repeated interaction, we believe that it also serves to highlight the importance of analyzing the different Stackelberg equilibria induced by different orders of play (which have long been known in economics to be potentially more desirable for the different players).
>
> In regard to the parameter $\mu$, you are correct that $\mu$ parametrizes the aggregate action of all the agents. We will clarify this when introducing the preliminaries.
>
> In regard to the underlying modeling assumption that the agents act as a group, we completely agree that it is important to understand what happens when agents do not act in coordination. We have actually thought of this as an interesting avenue of future work, and will gladly add some commentary on this in the discussion section.
>
> Please let us know if you have any remaining concerns or would like any more clarifications.

---

### Official Review · Reviewer_Ddzr · 2021-07-12

**Rating:** 6
**Confidence:** 5

**Summary:**

This paper studies a variant of strategic classification where there is not instantaneous interaction between the learner and the agents, but rather, there is a “slower” and a “faster” player (i.e., the whole interaction is governed by the frequencies according to which the learner and the agents update their decisions). This variant is closer to real-life settings, where for example the learner may have much more computational power compared to the agents. In their model, the agents are assumed to update their decisions at a fixed rate, and the learner gets to decide whether to be proactive (i.e., update his decisions slower than the agents) or reactive (i.e., update his decisions faster). After the learner announces the deployed model, the agents are broadly “rational”; to be more specific, they decide their reports based on no-regret algorithms (rather than the standard instantaneous best-response in prior work). The difference in timescales results in fact in a change of who the “leader” and who the “follower” is in the setting. The question that the paper addresses is what changes in terms of loss for the learner and the agents if there exists this timescale difference and whether there are settings where it is best for both the agents and the learner to always have one of the two.

The paper has two main results. First, it shows that if the learner is proactive (resp. reactive), then the learning dynamics of the system converge to the learner’s equilibrium (resp. agents’ equilibrium). Second, it shows that if the learner runs linear or logistic regression and the agents aim to maximize their predicted score, it is preferred by both the learner and the agents that the agents “lead” (i.e., that the learner is reactive). In terms of techniques, the paper uses variants of standard tools in Bandit Convex Optimization (BCO), assuming that everything is smooth enough (i.e., convex, smooth, Lipschitz).

**Ethical Concerns:**

No ethical concerns in my view.

**Limitations And Societal Impact:**

The limitations of the work are adequately addressed in Section 5. I also particularly enjoyed their observation about the meta-game that might be induced.

I think the potentially negative impact of the work should have been addressed, too. For example, you could include a paragraph citing some of the papers that have proved how strategic classification makes the disadvantaged groups worse off.

**Main Review:**

=====  Post Rebuttal  =====

I would like to thank the authors for addressing my comments and participating in the discussion. Please include the BCO discussion in a revision of the paper, together with the other comments you acknowledged. I'm increasing my score to a "weak accept" after our discussion where you stated that the surprising result on the order of play seems to also hold for more "adversarially" produced datapoints.

======================

Evaluation.
I think the paper’s strong trait is the observation of the different timescales, which is definitely novel and closer to reality than the prior models. The technical part of the paper is similar to prior work and in and of itself makes some assumptions on the form of the losses, the risks and the behavior of the agents (I do understand that the paper of [Dong et al., EC18] makes the same assumptions and proves the conditions under which they hold). The reason for my weak reject is that I think that the bound that you should have used for BCO part is the bound of [Bubeck et al., STOC17] and that the paper will benefit from a more clear writeup (I have included extended comments below).

Weaknesses.

-- The technical part of the paper is similar to prior work. And there is a second layer to that. The paper of [Dong et al., EC18] adapts the algorithm of [Flaxman et al., SODA05], which was indeed the first to achieve no regret in BCO, but ever since there have been much better algorithms. This is also acknowledged by [Dong et al., EC18]. I think that the paper should try to adapt the algorithm of [Bubeck et al., STOC17] which achieves O(poly(d) \sqrt{T}) regret, instead of the O(T^{¾}) one. By the way, I do understand the higher dependence in the dimension, I am just focused on dropping the dependence in T. If this is not possible, is there a fundamental barrier?

-- The two examples showcasing the counterintuitive preferred order of play are nice, but I think it would be more meaningful to consider more “adversarial” settings similar to prior work. Or, what happens if you have separable data (see the paper by [Ahmadi et al., EC21])? Are there any minimal conditions for the data generating process and the risk functions in order to get to the same conclusion that the agents’ leading is preferable for both?

-- I think that the paper could use some clarifications in the write-up. Examples: (and please correct me if I’m wrong in any of the following)

* I believe that you need to assume that the learner knows the agents’ frequencies in order to decide to be proactive/reactive. Why is it OK to assume such knowledge in real-life settings?
* You assume that the label/score y remains the same even after manipulation. Please state that clearly, since there are a lot of works recently where this is not the case.
* In 2.2 some details are loosely defined; for example, the “most rational decision” in Line 220 assumes a risk-averse, expected utility maximizer agent, right? Also, the no-regret behavior is not identical to the \mu_{BR}. This is indeed clarified in the first paragraph of page 6, but I was confused reading lines 217 -- 233 and then the definition of the no-regret behavior.
* What if for the faster player there is an “idle” round? In general, do the rounds for the updates need to “align”? I.e., the slow player plays once every 3 plays for the fast player.
* It is said early in the paper that the agents’ leading is a “preferred” order of play; please specify earlier that this preference is defined wrt to the accuracy loss for the learner, the loss of the agents, and their manipulation cost. Alternatively, you can formally define welfare in the beginning and argue wrt to it.

Strengths.

-- The model is very interesting, novel, and indeed it models much better the interactions in strategic classification settings. Frankly, the idea of studying different timescales for the learner and the agents is one of the missing pieces in the strategic classification literature and the main selling point of the paper.

-- The observation that there are settings where *both* the agents and the learner prefer that the agents’ lead is indeed surprising and counterintuitive. The paper does a good job of explaining also the relationship with zero-sum games.

Additional comments.

-- Missing related works: [Braverman & Garg, FORC20], [Ghalme et al., ICML21], [Levanon & Rosenfeld, ICML21], [Jagadeesan et al., ICML21], [Ahmadi et al., EC21], [Bechavod et al., arXiv21]

-- Line 180: what is m? Why is this not equal to d?

-- Line 315: what is D_{t+1}?


**Time Spent Reviewing:**

6-7 hrs

---

> ### Author Response · Authors · 2021-08-10
> **Response to Reviewer Ddzr**
>
> Thank you for your comments and constructive feedback. In what follows we address the weaknesses pointed out in the review.
>
> (Choice of BCO algorithm)
>
> We agree that there has been a lot of subsequent work in BCO with various improvements relative to the work of Flaxman et al., and we will gladly add some more discussion on it. That said, we do not feel that the algorithm of Bubeck et al. is objectively better than the proposal of Flaxman et al.; the choice to prefer the dependence on $T$ over the dependence on $d$ is arguably subjective. As you point out, the dependence on $T$ goes from $T^{3/4}$ to $\sqrt{T}$, but the dependence on $d$ becomes *far* worse (and cannot simply be ignored in the big-O notation): it goes from $\sqrt{d}$ to $d^{9.5}$. For example, for $d=2$, $\sqrt{T} d^{9.5} < T^{3/4}\sqrt{d}$ only when $T>2^{36}$, which exceeds 68 billion epochs. In addition, the algorithm of Flaxman et al. is arguably much simpler and more interpretable. We are not saying the algorithm of Bubeck et al. is necessarily inferior---that $\sqrt(T)$ regret is possible is a very interesting theoretical finding---but practically it does not seem to improve upon the Flaxman et al. solution in any strict sense. Further, we note that Dong et al. do not refer to the algorithm of Bubeck et al. as “better”; they merely mention it as a possible alternative that is better in one sense and worse in another. We hope that you agree that there is no strict overall ordering between the two algorithms.
>
> More importantly, this choice of BCO algorithm is secondary to our list of technical and conceptual contributions. Our main contributions are the introduction of update frequencies, the no-regret model of agent behavior in this context, the possibility of equilibria with different orders of play, and the counterintuitive observation that the agents’ equilibrium might have advantages over the decision-maker’s equilibrium. These are the aspects of the paper that we emphasize in our exposition; we do not state the BCO rate as our original contribution and we mention it only very briefly in the paper. We use the BCO algorithm only as a proof of concept that simple, standard algorithms can find equilibria despite assuming only no-regret behavior on the agent side, something which does not follow from existing work in the literature. We do not think there is a fundamental barrier in studying the same extension with the Bubeck et al. algorithm, but it is certainly technically more involved.
>
> We hope that you will consider having a fresh look at our paper with our intended contributions in mind.
>
> (Conditions for preferred order of play)
>
> We have thought about more general conditions for order-of-play preferences and found this to be a non-trivial question which merits an independent investigation in future work. The two stylized examples in Section 4 are meant to illustrate the benefits of a more general framework that takes into account update frequencies; we hope to look into further conditions in the future. Could you clarify what you mean by “more adversarial settings”? We would be happy to look into this.
>
> (Clarifications)
>
> We will incorporate additional discussion to address the points you raised in the bullets. A summary of the clarifications:
> 1. Indeed, in line 192 we say that the decision-maker is aware of the agents’ timescale. Big tech companies generally employ monitoring systems that detect distribution shift; at a high level, the rate of distribution shift can be thought of as the rate of agents’ adaptation. Since we only initiate this study we consider a simplified setting with a known timescale, however we agree that it is an interesting question to understand settings where the update frequency must be empirically inferred.
> 2. Our general theoretical treatment does not require the label/score to remain the same after manipulation. This is only true in our illustrative examples, which we will clarify in our informal discussion of these examples in the introduction.
> 3. We apologize for the confusion due to the discussion in lines 217-233; these paragraphs were only meant to help with intuition, not to be treated as technical preliminaries. Formally we assume no-regret behavior. We will clarify the distinction between formal assumptions and high-level intuitive discussions.
> 4. Formally, idle rounds can be thought of as repeatedly playing the same action, hence idle rounds are fine (as long as in the limit the relevant conditions hold, e.g. no-regret behavior).
> 5. We will clarify that the “preferred” order of play refers to a narrow set of criteria early on.
>
> (Additional comments)
>
> Thank you for pointing out additional related work, we will add the references in the next version of our paper. (Note that some of these papers appeared online only after the submission deadline.)
>
> In general the vector parameterizing the agent distribution $\mu$ need not live in the same space as the model $\theta$, hence $m$ need not be the same as $d$. As a toy example, $\mu$ could just be a single scalar and $\mathcal{P}(\mu) = \mu \mathcal{P}_0$ could be a rescaling of a base distribution. The model $\theta$ can still live in $\mathbb{R}^d$.
>
> We will clarify that $D_{t+1}$ is just an arbitrary map (we are basically defining it in line 315).
>
> Seeing that we can clarify the points you raised without significant rewriting of the paper, and that the choice of the BCO algorithm is secondary to our contributions, we sincerely hope you will reconsider your score with our principal contributions in mind.

---

> > ### Comment · Reviewer_Ddzr · 2021-08-25
> > **regarding more "adversarial settings"**
> >
> > Thank you for your detailed response!
> > 1) Agree to disagree regarding the algorithm of Bubeck et al. The Flaxman et al. may be more practical, but I think that the Bubeck et al. one is considered the state-of-the-art in the ML theory literature. I agree that this may be secondary to the points you are trying to make, but my point was on the technical novelty of the present work. In some sense, I wanted to argue that the techniques presented may not translate to the BCO algorithm of Bubeck et al., but I realize that I didn't communicate this clearly. If you have thoughts on this, I would greatly appreciate them though.
> > 2) Regarding more “adversarial” settings: so currently in your examples there is a predefined relationship between the features and the labels/scores. But settings like Dong et al., Chen et al. and Ahmadi et al. do not pose such assumptions. So to clarify: assume you are only given separable data from the agents (i.e., not coming from a specific data generating process, but still separable by a margin), can you still make similar claims regarding the reversed order of play?
> > 3) I think the only paper that hadn't appeared prior to the deadline is the one by Jagadeesan et al.

---

> > > ### Author Response · Authors · 2021-08-26
> > > **Further Discussion and Results**
> > >
> > > Thank you for following up and engaging in the discussion, we appreciate your comments and your time.
> > >
> > > (BCO algorithm)
> > >
> > > We are happy to add more discussion on how our analysis can be extended to the Bubeck et al. algorithm, an overview of which we present now. In the following we are referring to the steps in Algorithm 1 in the arxiv version of the paper [https://arxiv.org/pdf/1607.03084.pdf]. As in our submission, the decision-maker would use $L(\bar \mu_t, \theta_t)$ as the zeroth-order feedback in step 12 ($\theta_t$ is our notation for $x_t$ in the Bubeck et al. paper). Per the same Lipschitz argument as in our proof, this feedback can be written as $L(\mu_{\text{BR}}(\theta_t), \theta_t) + \text{err}\_t$, where $|\text{err}\_t| \leq \beta\_\mu \|\bar \mu\_t - \mu\_{\text{BR}}(\theta\_t)\|\_2$. In step 14 $\text{err}\_t$ gets multiplied by a bounded factor, hence in steps 16 and 17 if we let $\tau\rightarrow \infty$, this error term vanishes by the assumption of no-regret behavior (as in our proof). Therefore, at least asymptotically in $\tau$ we do not see any issue with an analogous adaptation of the Bubeck et al. algorithm that would yield a $\sqrt(T)$ rate.
> > >
> > > Overall, we agree that the Bubeck et al. paper has made impressive theoretical breakthroughs in the BCO literature. However, looking at the papers that cite it, the algorithm is widely acknowledged to have drawbacks both theoretically (for its dimension dependence) and practically. For example, experts in the field (Agarwal, Luo, Neyshabur, Schapire, “Corralling a Band of Bandit Algorithms”) say that it “[makes] some important progress [...] but unfortunately with very complicated and impractical algorithms”. Other papers that make similar comments include [1,2,3,4] below. Again, we are really not claiming that the Flaxman et al. algorithm is better, but we would appreciate feedback on why our comment about the dimension dependence of the Bubeck et al. algorithm can be dismissed when comparing the two algorithms objectively. Our main goal was to show how a decision-maker using simple, standard, algorithms could push no-regret learners towards different equilibria in strategic classification.
> > >
> > >
> > > (Adversarial Settings)
> > >
> > > Thank you for clarifying your question about “adversarial” settings. Indeed, our result in the classification setting holds more generally (we used a specific data-generating model to make some of the steps in the argument more explicit).
> > >
> > > To be specific, suppose there is a non-manipulated dataset of $n$ fixed observations, $(x\_1,y\_1),...,(x\_n,y\_n)$, where the labels are binary, $y\_i\in\{0,1\}$. Assume the rest of the setup as in our paper (agents with $y_i=0$ have a fixed budget for manipulation, decision-maker runs ERM with the logistic loss on this dataset, etc). Then, *both the decision-maker and the agents prefer the agents to lead*. This claim follows by almost the same argument as in the Supplement. In particular, the argument for why the decision-maker prefers to follow remains unchanged. For the agents, we apply a similar argument as in our proof: the idea is to show that if the agents play $\frac{\theta\_{\text{SE}}}{||\theta\_{\text{SE}}||\_2 }B$, then $(\theta\_{\text{SE}}, \mu\_{\text{BR}}(\theta\_{\text{SE}})) = (\theta\_{\text{BR}}(\frac{\theta\_{\text{SE}}}{||\theta\_{\text{SE}}||\_2 }B), \frac{\theta\_{\text{SE}}}{||\theta\_{\text{SE}}||\_2 }B)$. This equality follows by observing that $L(\mu,\theta)$ is convex in $\theta$, and by showing that $\nabla\_\theta L(\frac{\theta\_{\text{SE}}}{||\theta\_{\text{SE}}||\_2 }B, \theta\_{\text{SE}}) = 0$ by the definition of $\theta\_{\text{SE}}$. Since there exists an action under which the agents can achieve the same loss when they lead, as when they’re at the decision-maker’s equilibrium, that implies that their loss can only be lower (or equal) at $\mu\_{\text{SE}}$. We would be happy to highlight this more general statement and proof in the paper.
> > >
> > >
> > >
> > > Thank you for your time and continued involvement, we hope that this additional discussion addresses your concerns.
> > >
> > > [1] Saha, Natarajan, Netrapalli, and Jain; Optimal regret algorithm for Pseudo-1d Bandit Convex, 2021
> > >
> > > [2] Garber and Kretzu; Improved Regret Bounds for Projection-free Bandit Convex Optimization, 2019
> > >
> > > [3] Farina, Schmucker, and Sandholm; Bandit Linear Optimization for Sequential Decision Making and Extensive-Form Games, 2021
> > >
> > > [4] Tatarenko and Kamgarpour; Minimizing Regret of Bandit Online Optimization in Unconstrained Action Spaces, 2020

---

> ### Author Response · Authors · 2021-09-01
> **Feedback**
>
> Thanks for your hard work reviewing. We just wanted to reach out and see if you had any comments or questions after our response given that the discussion period is ending. Specifically, we believe that we gave a significant generalization of our logistic regression result to the “adversarial” setting as you described it. We also hope that we have managed to illustrate that our techniques do indeed generalize to the Bubeck et al. BCO algorithm.
>
> We hope that with these two points we have addressed your main concerns.

---

### Official Review · Reviewer_nkip · 2021-07-12

**Rating:** 6
**Confidence:** 3

**Summary:**

The paper argues for a more general view on strategic classification of objects modified by adversaries. In the standard view, the designer of the classifier is assumed to be the leader and the classifier is optimized with respect to a best response form the data generator to its parameters. This paper argues that in some situations, it may be possible and desirable to reverse the roles and force the rational attacker to create objects first and optimize them with respect to the expected best response classifier. The paper shows how different update frequencies of individual players using no regret and gradient descent learning can lead to convergence to the equilibria in either assignment of the roles. Furthermore, the paper shows that for linear and logistic regression, each of the role assignments can be desirable for each of the players depending on parameters, such as the magnitude of the allowed perturbation by the attacker.


**Limitations And Societal Impact:**

The authors should analyse more carefully the limitations of their formal model (i.e., the data not changing) and the empirical results achieved (i.e., how realistic the parameters that lead to the difference are). For societal impact, the authors do not discuss it, however, I do not consider the potential negative impact of this work to be substantially larger than in case of any other work on adversarial classification.

**Main Review:**

Originality: I agree with the first parts of the paper that claim that most existing work considers the classifier to be the leader in a Stackelberg game and I am not aware of any work that would reverse the roles. The idea to reverse the roles is not entirely intuitive and therefore sufficiently novel. Also, showing (in highly idealized setting) that the equilibria corresponding to the different orders can be induced by the speed of updating is interesting and novel.

Technical Quality: The main statements of the paper intuitively make sense. I did not carefully check the detailed proofs that are provided in a lengthy appendix. The main text is rather formal and reasonably clear, but some things are missing. For example, it makes an assumption that all best responses and equilibria are unique without any justification. Some of the notation is not properly introduced. For example, a crucial parameter “diam()” used in Theorem 3.1 is undefined, I / I_d in section 4 is undefined, ->_p for “convergence in probability” should also be defined more exactly.

Clarity: The main idea of the role reversal is clear. However, it is not enough on its own for a neurips publication and it is not very clear how some parts of the paper support it. The paper would, in my opinion, be substantially clearer if it introduced a comprehensive detailed description of a realistic problem, that would require the kind of analysis the paper promotes.  In the presented formal model, there is no change of data and the only thing evolving in time are the player’s strategies. Assuming rational players, I do not see much reason for having gradual updating on different time scales. In the given example with spam, the spammer would just craft the hardest to detect template and start using it, while the classifier would create the best classifier for it. If the equilibrium is unique as assumed, the spammers would not have any incentive to change their strategies anymore. Since the data do not change and the equilibria are unique, it is not clear why it is useful to analyze the process of adaptation of the classifier on the level of gradient descent.
If the gradual updating is somehow justified, the deeper, less trivial analysis in the paper is on the more standard form of the classifier in the role of the leader, which is unexpected after the introduction. There, the result in Theorem 3.1. is interesting and non-trivial. On the other hand, the analysis of the equilibrium for the reversed roles basically assumes that the classifier is trained to full convergence, which is much less interesting and novel.

Significance: The paper presents an interesting theoretical idea. However, it does not sufficiently argue for its practical impact. The paper shows some parameter settings, where the players’ payoffs in the different equilibria differ. However, the ability of the attacker needs to be quite extreme (over 1.5 change in data with unit variance (if I interpret the undefined I correctly). A more compelling discussion of practical impact would be necessary to consider the paper significant practically. If we focus on the theory, the paper may inspire future work that will consider the agent modifying the data as the leader.  It sufficiently shows that the case exists, and could potentially have some implications on players strategies, depending on parameters. However, I am not convinced that the paper provides sufficiently novel and interesting tools for deeper analysis of this problem to warrant a publication at NeurIPS.

Summary: This paper presents a very interesting idea of role reversal and argues well for the need to analyse different adaptation speed of the players in adversarial classification. However, I do not think it provides sufficient evidence for its practical impact or sufficiently general, deep and reusable theoretical framework that would make its future analysis substantially simpler.


After the discussion with the authors, they managed to resolve some of my confusions and I was happy to increase my score. While uniqueness would be a problem in non-linear classification or when assuming mixed strategies, I am ok with it in case of linear classifiers and pure strategies. We have also resolved the problem of changing data. I hope a future version of the paper will be much clearer about which agent has access to which information, but I think my main concern is resolved by the realisation that z is likely only a sample of the data that can be different in each round, even though it comes form the same static distribution, generally unknown to the players.

The concern that was not fully resolved is that the more interesting results with regret minimisation are for the more standard setup of the classifier as the leader and the reversed theorem essentially assumes a best-responding opponent, which is much less novel.

**Time Spent Reviewing:**

6

---

> ### Author Response · Authors · 2021-08-10
> **Response to Reviewer nkip**
>
> Thank you for your time spent reviewing our paper. We appreciate your comments and hopefully we can clear up your concerns.
>
> (Uniqueness of best responses and equilibria)
>
> The uniqueness of best responses is a standard, often implicit, assumption in the literature on strategic classification and game theory more broadly. Uniqueness of best responses follows when a player’s loss is strictly convex when the other players' actions are fixed. Uniqueness of Stackelberg equilibria is also rather common in the literature, as the Stackelberg risk is often assumed to be convex (see, e.g., [13, 23, 32, 33]).
>
> (Undefined notation)
>
> We apologize for not defining some notation, we used standard notation in the literature. By "diam" we refer to the diameter of a set and $I$ and $I_d$ are identity matrices. We are happy to clarify this in a revision.
>
> (There is no change of data and the only thing evolving in time are the player’s strategies.)
>
> The agents’ data *does* in fact evolve with time; the distribution $\mathcal{P}(\mu)$ denotes the agents’ data distribution after they take action $\mu$. In the spam example, the agents’ emails are the data, and the features of the emails change depending on the action $\mu$. Similarly, in our illustrative examples on logistic and linear regression the data the decision-maker sees evolves over time as a function of the agents’ strategies.
>
> (Assuming rational players, I do not see much reason for having gradual updating on different time scales. In the given example with spam, the spammer would just craft the hardest to detect template and start using it, while the classifier would create the best classifier for it.)
>
> Designing a hard spam email and learning a good spam classifier require iterative learning. If a spammer wants to find the “hardest possible” email for the spam filter, they have to repeatedly interact with the spam filter to learn what features make an email hard to filter out. The spammer cannot know what a difficult-to-detect spam email looks like before any interaction with the filter. That is why the two players have to gradually adapt to one another in order to reach an equilibrium.
>
> Understanding the dynamics that arise out of gradual adaptations by players in games is a research area that has gained a lot of traction in recent years (see, e.g., [6,14,15, 24, 29]) and has a long history in game theory going back to at least Rosen (1965). Our results expand upon those results by showing how simply changing the update frequencies of the local adaptations can lead to different equilibria. The literature has in the past focused either on simultaneous updates (e.g., [6, 24 ,29]) or assumed a fixed order-of-play (e.g., [14,15]).
>
> (On the other hand, the analysis of the equilibrium for the reversed roles [Theorem 3.2] basically assumes that the classifier is trained to full convergence, which is much less interesting and novel.)
>
> We would like to push back respectfully on this comment, because while we do not give an explicit rate of convergence in Theorem 3.2, we are not aware of any work that proves convergence to the agents’ equilibrium or that assumes only no-regret agent behavior as opposed to best-responding agents. While the result may seem simple, it is a consequence of our more complex and realistic model of strategic classification that incorporates timescales. Coupled with our examples on logistic and linear regression, Theorem 3.2 gives simple and realistic conditions under which new (and potentially more desired) equilibria can be achieved: a) the decision-maker updates the model at high frequencies and b) the agents are no-regret learners. The result is asymptotic simply because no-regret learning is a fairly weak assumption on the agents’ behavior. For a finite-time rate in the style of Theorem 3.1 we would have to assume a specific learning algorithm used by the agents, which we believe to be too strong an assumption to make.
>
> Thank you for your consideration of our response! Please let us know if you have any remaining concerns.

---

> > ### Comment · Reviewer_nkip · 2021-08-16
> > **My main concerns remain unresolved**
> >
> >
> > The rebuttal has answered some of my concerns, but unfortunately not the main ones regarding novelty and real-world significance of the results.
> >
> > (Uniqueness of best responses and equilibria)
> >
> > Thank you for the clarification. I was thinking about the context of mixed strategy Stackelberg equilibria. For pure strategy equilibria and linear classifiers, I do not have a problem with the uniqueness.
> >
> > (There is no change of data and the only thing evolving in time are the player’s strategies.)
> >
> > Line 176: “Throughout we denote by z = (x, y) the feature, label pairs corresponding to the strategic agents’ data.” The data stays the same z throughout the whole paper and never changes.
> >
> > Based on this, the definition of the losses and the optimization problems, I assumed that the agents have access to z and know the naturally defined utility functions of adversarial classification. Therefore, they can just compute the equilibria and do not have to iterate. If this is not the case, the paper should be much more clear about which player has access to which information at what point of the game.
> >
> > After reading the rebuttal, I tried to decipher this on my own, even though it is not an easy task. Let’s assume the spam filtering example and the decision maker as the leader. In order to estimate the loss with respect to the best-responding agent, she needs to know the dataset z and the loss function of the attacker. I do not see how she could make multiple iterations of gradient descent w.r.t BR without deploying the model and without access to this information. If the decision maker is the follower, she could possibly react to the number of reported spam messages by the user. She could also use the sample of transformed z to perform a sampled gradient update. Still, it looks like at least the leader needs full access to the dataset and the losses and hence can compute the equilibrium without real-world iterations.
> >
> > (On the other hand, the analysis of the equilibrium for the reversed roles [Theorem 3.2] basically assumes that the classifier is trained to full convergence, which is much less interesting and novel.)
> >
> > I assume you mean Section 3.2., since there is no Theorem 3.2. in the paper. As I read it, it says that if a GD-BR dynamics converges, it has to converge to a Stackelberg equilibrium. As the authors write on lines 132-144, the dynamics where one of the players plays the exact best response has been explored before, so I am still not sure about the novelty of this part. As I wrote before, the no-regret learning part is in my opinion novel, but that analysis was used only for the classifier as the leader.

---

> > > ### Author Response · Authors · 2021-08-17
> > > **Further Clarifications**
> > >
> > > Thank you for following up and engaging in the discussion, we appreciate your comments and your time. Based on the response we feel there are still important misunderstandings of our work and we would like to clarify the points raised.
> > >
> > > (There is no change of data and the only thing evolving in time are the player’s strategies.)
> > >
> > > We believe that the misunderstanding about changing data stems from a misinterpretation of our notation; the data that the decision-maker observes *does* change. The purpose of the sentence you copied is just to say that $z$ variables are used to denote the tuple $(x,y)$. After the agents play $\mu_t$ at time $t$, the decision-maker observes distribution $\mathcal{P}(\mu_t)$; we use $z$ only as a dummy variable when we want to denote a sample from $\mathcal{P}(\mu_t)$, $z\sim\mathcal{P}(\mu_t)$. We can also write $z_t\sim \mathcal{P}(\mu_t)$ and this would make no formal difference. In short, the decision-maker observes $\mathcal{P}(\mu_t)$ in epoch $t$, and this distribution varies with time because $\mu_t$ varies with time.
> > >
> > > The decision-maker and the agents only know *their own* loss function, and they observe the actions of the other player over time. This is a point we can easily clarify in the writeup. In particular, the decision-maker observes the data corresponding to the strategic agents, and the agents observe the current model deployed by the decision-maker. As a result, they cannot compute the equilibrium “offline”, before any interaction.
> > >
> > > Specifically, the leader need not know both losses, as suggested. One key insight is that the leader can use a zeroth-order gradient estimator to *approximate* the gradient of their Stackelberg risk even when the agents’ are only playing no-regret. They can only approximate the gradient because the agents are not at their best response and because the leader does not have access to the agents’ loss. By using the zeroth-order estimator, convergence to equilibrium is achieved without ever knowing the follower’s loss.
> > >
> > > (Novelty of Theorem 3.5/Section 3.2)
> > >
> > > Yes, we meant Theorem 3.5/Section 3.2, apologies for the confusion. Theorem 3.5 *does* indeed require no-regret learning of strategic agents as an assumption (A2). No player in this theorem is assumed to be playing exact best response. The decision-maker is using, say, gradient descent while the agents use a no-regret algorithm. Thus, this theorem implies asymptotic convergence of the dynamics of *gradient descent* and *no-regret* players to the agents’ equilibrium under the assumption that the decision-maker makes frequent updates. No work mentioned in lines 132-144 proves convergence of these dynamics. More importantly, to the best of our knowledge, there is no known result in strategic classification that shows convergence to a Stackelberg equilibrium where the strategic agents lead in the game. This result emerges from our new model with relative update frequencies.
> > >
> > > Again, we appreciate your time and your continued engagement. We hope that we were able to clarify the novelty of our work, and that you can re-evaluate the paper with these points in mind.

---

> > > > ### Comment · Reviewer_nkip · 2021-08-24
> > > > **I am still not fully satisfied, but I am happy to increase my score and change the recommendation.**
> > > >
> > > > Thank you for the clarification. I think I now understand that the training data is sampled in each iteration from the distribution selected by the data generator for that round and if the classifier adapts faster, it can make its gradient updates w.r.t. this data and their correct labels. If the data generator is assumed to move faster, she generates new samples from her current distribution and updates it based on a no-regret dynamics. When I am double-checking it now, it is somehow written there, but it was not clear to me for the first reading.
> > > >
> > > >
> > > > As for the Theorem 3.5., I might be misinterpreting the double infinite limit. Without further explanation, I read it so that for each T, you need to go with \tau to infinity. That is what I meant by computing the best response, regardless whether you assume gradient or no-regret dynamic. Even the proof seems to be going that way. Hence, I am not still not convinced that the theorem is really saying something new.
> > > >
> > > > Still, I agree that a large part of my confusion might be reduced by relatively minor improvements in writing and I am happy to increase my score.

---

> > > > > ### Author Response · Authors · 2021-08-26
> > > > > **Response**
> > > > >
> > > > > Thank you for taking the time to respond to our comments and for your continued engagement, we really appreciate the time and effort you’ve put in.
> > > > >
> > > > > Per your comment about Theorem 3.5, thank you for clarifying. You are correct that the theorem operates in the limit where the decision-maker is essentially converging to the agents’ best-response. We agree that, technically, the proof of the theorem is relatively simple.
> > > > >
> > > > > *Conceptually*, however, we believe this theorem is saying something rather important: by abandoning the usual assumption of best-responding *agents* (and replacing this with no-regret learners), we show how just by changing their algorithm and update frequency, the decision-maker can drive the strategic agents towards a different equilibrium that has not been explored in the strategic classification literature. As we show, these equilibria can be more desirable than those previously studied.
> > > > >
> > > > > Thank you again for your comments and continued involvement.

---

### Decision · Program_Chairs · 2021-09-27

**Decision:**

Accept (Poster)

**Comment:**

While the scores appear marginal, I still recommend this paper for acceptance.  The authors responded well to some points and some issues raised were found not to be much of a problem.  It is clear that some work is needed on the writeup in line with the reviewers' comments (e.g., contrasting with the Ball paper), but it is very reasonable to assume this can be done before the camera ready.

While we certainly can't fault the authors for not having a high enough reference count, there are a few other things that could be expanded.  For example, there is a paragraph on learning in Stackelberg games, but that literature (not necessarily restricted to the classification context) goes back to 2009 if not further, and not just for zero-sum games.  This seems worth discussing.  There is quite a bit of additional recent work on learning in non-zero-sum strategic classification settings, including relationships to mechanism design (which is itself a Stackelberg model), as well; see e.g. "Incentive-Aware PAC Learning" and other work cited therein.  To my knowledge, none of this makes the authors' work less valuable -- if anything, it makes it more so.  Finally, there is work in the economics literature on justifying Stackelberg models/outcomes through different degrees of patience between the players, e.g., "Reputation and Dynamic Stackelberg Leadership in Infinitely Repeated Games" -- it is not quite clear to me what the relationship is to update frequencies, but perhaps something interesting can be said.